# Adults with more severe psychopathy in the community show increased social discounting

Naomi Nero [1] ✉, Marla Dressel [1], Paige Amormino [1], Paige Freeburg [2], Melinda Somers [1], Lindsay Poplinski[3], Katie Duhamel [3], Viviana Alvarez-Toro [3] & Abigail A. Marsh [1]

Psychopathy is a personality construct characterized by boldness, disinhibition, insensitivity to others' suffering or distress, and persistent engagement in behaviors that harm others. These combined features suggest that highly psychopathic people may place much less subjective weight on others' outcomes relative to their own. We therefore assessed social discounting, which indexes how the subjective value of others' outcomes declines as a function of social distance, in a demographically diverse community sample of very-high psychopathy adults (above the 95th percentile of TriPM scorers; $n = 288$), as well as a sample of demographically similar controls ($n = 427$), who also reported antisocial and criminal behavior. Results show robust increases in social discounting as psychopathy increases ($p < 0.001$), and that reduced subjective valuation of others' outcomes partially mediates the group differences in antisocial behavior ($p = 0.018$). These insights emphasize the importance of understanding how psychopathic traits manifest in the community and underscore how diminished valuation of others' outcomes represents an important mechanism driving maladaptive behaviors.

Psychopathy is among the strongest dispositional predictors of antisocial behaviors that cause others distress, suffering, or harm, ranging from lying, theft, and manipulativeness to violence and criminal offending[1–3]. Psychopathy is particularly closely linked to instrumental aggression, or intentionally harming others for personal gain[4–7]. Core affective features of psychopathy include callousness and uncaring, which reflect relative insensitivity to others' suffering or distress[2,8]. That causing instrumental harm and insensitivity to others' suffering are core features of psychopathy suggests psychopathy may be characterized by reduced subjective valuation of others' welfare. That is, highly psychopathic people may harm others and fail to care if they suffer because they place little subjective weight on others' outcomes relative to their own. But no prior study has quantified the relationship between psychopathy and the subjective valuation of others' welfare, or whether this variable can account for affective and behavioral features of clinically significant psychopathy.

Psychopathy is generally agreed to vary along a spectrum in forensic, clinical, and community samples[9–11]. Features of psychopathy include callousness, boldness, and impulsivity[7,12], with 1-5% of the population exhibiting clinically significant levels of these traits[4,7,13–15]. People with psychopathic traits consistently exhibit higher levels of antisocial and criminal behavior[7]. Previous research has suggested a variety of factors that

may drive antisociality in psychopathy. One is deficits in learning from punishment or threat[16,17], for example, impaired response reversal and passive avoidance learning in response to aversive stimuli[18,19]. These patterns may in part reflect low fear responding in psychopathy[20–25], which may result from atypical patterns of neurodevelopment that render punishments less effective as deterrents[26–29]. Antisociality in psychopathy and other disorders may also, in part, reflect deficits in executive functions like cognitive control[2,30–32], as well as social deficits, such as reduced empathic and perspective-taking abilities[7,33–37], and decreased affiliative motivation[38,39]. Accordingly, psychopathy has been linked to reduced prosocial behavior using tasks such as the dictator game and donation paradigms[40–43]. However, little research has considered whether these features reflect people with psychopathy assigning less subjective value to others' outcomes.

The social discounting task was developed to quantify the subjective valuation of others' welfare. The task is modeled on temporal discounting tasks, which index decreases in the subjective value of rewards as a function of delay. The social discounting task instead indexes decreases in the subjective value of rewards as a function of the social distance of the person the reward is shared with. In this task, respondents make choices to keep resources or share them with real people of increasing social distance[44–46], allowing the subjective value of outcomes for the self ($N = 0$) versus others at

[1]Department of Psychology, Georgetown University, Washington, DC, USA. [2]Department of Psychology, Yale University, New Haven, Connecticut, USA. [3]St. Elizabeths Hospital, DC Department of Behavioral Health, Washington, DC, USA. ✉e-mail: nn444@georgetown.edu

varying social distances ($N = 1$–100) to be calculated. Declines in the subjective value of a reward follow a hyperbolic function across increasing social distances [46]. Advantages of this task over other paradigms involving resource allocation, such as the dictator game and donation games, include the use of multiple trials per recipient, as well as decisions for multiple real recipients who vary in social closeness (rather than a single anonymous stranger or abstract organization). This structure enables the calculation of a reliable value that can be interpreted as representing the subjective valuation of actual others' welfare.

Because the majority of actual prosocial behavior is aimed at benefiting close others rather than strangers [47–49], this task also benefits from increased ecological validity relative to other commonly-used paradigms such as the dictator game and social value orientation task (SVO) that focus on generosity toward a single anonymous stranger and which were created to assess other constructs (for example, individualistic versus competitive, altruistic, or cooperative outcomes in the case of the SVO). As a result of these task features, the social discounting task has higher predictive validity than other prosocial tasks, including the SVO and dictator game [50] or self-report measures [51,52]. Furthermore, neuroimaging and behavioral research support the conclusion that choices during the task reflect variation in the subjective valuation of others' welfare rather than effortful suppression of selfish responses [52].

Thus, the current study used the social discounting task to assess how the subjective valuation of others' welfare varies as a function of psychopathy. Three prior studies using undergraduate samples have altered features of the social discounting task to assess, respectively, communion [53,54] and generosity for single versus multiple people [55] in psychopathy. These studies found mixed results, with two studies finding decreased generosity was associated with psychopathy [53,55] and the third finding no relationship between psychopathy and discounting [54]. Antisocial behaviors more broadly, including self-reported texting while driving [56] and adolescent externalizing symptoms [57] have been linked to increased social discounting. None of these studies, however, has found a link between psychopathy and social discounting using the standard version of the social discounting task that aims to quantify subjective valuation of others' welfare, or assessed social discounting in a sample with clinically significant psychopathic traits.

Sampling is a persistent challenge in studying neurocognitive features of psychopathy. High-psychopathy samples are often recruited from forensic or psychiatric institutions, carrying challenges related to constrained recruitment and testing opportunities, non-representative samples, and difficulties determining whether neurocognitive impairments are core features of psychopathy or the result of institutionalization [58]. Disagreement persists about whether research in university students and other community samples with low average levels of psychopathy can be extrapolated to understanding clinically significant psychopathy [59,60]. And studies of so-called "successful psychopathy" that recruit from industrial settings or unemployment agencies tend to have small sample sizes [61–64]. Thus, little research to date has been conducted in well-powered non-institutionalized high-psychopathy samples recruited from the community.

We assessed social discounting in very high-psychopathy adults (above the 95th percentile of TriPM scorers) recruited from the community and demographically similar controls. Following prior work linking reduced social discounting to highly prosocial phenotypes [52], we predicted psychopathy would be associated with increased social discounting. We also predicted that the association between psychopathy and antisociality would be partly mediated by reductions in the subjective valuation of others' welfare as psychopathy increases. We also considered potential effects of age, gender, socio-economic status (household income), and cognitive intelligence, given prior evidence that increases in these variables are reliably associated with increased generosity [50,65–69]. Additionally, in light of disagreements about whether findings in high-psychopathy samples can be generalized to the general population, we conducted both group-based analyses comparing high psychopathy and control groups, continuous analyses across the full sample, and, where relevant, separate analyses within the high-psychopathy and general-population samples.

## Methods

This study was approved by the Georgetown University Institutional Review Board (ID#: 0000193). All participants provided informed written consent before the commencement of testing and were informed that the confidentiality of their responses was protected by a Certificate of Confidentiality from the National Institutes of Health. The study was not pre-registered. However, the authors completed and presented results utilizing a multiverse approach by using two common methods of analyzing discounting tasks (logk and AUC) while also examining group differences along with psychopathy continuously. Primary analyses examining the relationship between psychopathy and logk, and whether logk moderates or mediates the relationship between psychopathy and antisocial behavior, are presented in the main manuscript. In the Supplementary Information file, we report parallel analyses using AUC, an alternative discounting metric (Tables S18-26). All data and analysis code are publicly available [70].

### Participants

A total of 727 participants took part in this study, a sample size determined using the effect size generated from a recent meta-analysis [71], that found this sample would yield >80% power to identify group differences in social discounting at a statistical threshold of $p < 0.05$. Participants included a unique community sample of 366 very high-psychopathy participants recruited through the 501(c)(3) non-profit organization *Psychopathy Is* (now The Society for the Prevention of Disorders of Aggression, https://www.disordersofaggression.org), which provides information and resources for individuals and families affected by psychopathy and related disorders. Visitors can complete screening tests on the website, including the Triarchic Psychopathy Measure (TriPM) [12], a 58-item self-report measure that assesses three psychopathy subscales: boldness, meanness, and disinhibition (to protect visitors' privacy, no data are collected about participants or their scores on this measure by the website). Participants who receive TriPM scores in the top 5% of American adults of their gender [4] receive information about diagnostic and treatment options, as well as a link to take part in research upon providing their contact information, age, gender, and country of residence. We invited the 1242 respondents who had provided this information as of 11/13/23, were 18 or older, and indicated they reside in the United States to complete the study. Of these, 464 confirmed interest in participating in this study. We continued online recruitment until we achieved our intended sample of high-psychopathy participants. In addition, 361 control participants were recruited through CloudResearch. These participants completed identical measures. Controls were recruited to approximately match the high-psychopathy participants in terms of gender, age range, and race/ethnicity. Upon completion of the online Qualtrics survey, participants were compensated $15.

Prior to conducting group-based analyses, 78 participants whose TriPM scores fell below the estimated 95th percentile for their gender (Male=105; F or O = 91) were removed from the high-psychopathy group and reassigned to the control group. Cutoffs were derived from percentiles calculated using scores from a quasi-representative sample of U.S. adults who completed the TriPM [4]. In addition, 10 controls who scored above the cutoff scores for their gender were reassigned to the high psychopathy group (results were consistent when data were re-analyzed after dropping all reassigned participants; see Supplementary Information; Table S27-S48). Finally, 12 participants who failed two or more of four attention checks ($n = 10$ high-psychopathy and 2 controls) were excluded. Thus, our final sample of 715 included 288 high-psychopathy participants and 427 controls who were between 18-79 years old ($M = 36.7$ years; gender and race/ethnicity reported in Table 1).

Psychopathy was confirmed in our high-psychopathy group via follow-up screening using the Psychopathy Checklist: Screening Version (PCL-SV) [72], which was administered to 15% of 288 high-psychopathy participants ($n = 44$). The PCL-SV is a semi-structured interview-based assessment considered to be a reliable and valid measure of psychopathy, resulting in a total score that is further broken into a Factor 1 and Factor 2 score [73,74]. Adapted from the Psychopathy Checklist Revised (PCL-R), the

**Table 1 | Characteristics of high-psychopathy and control participants**

| | Control | High Psychopathy | p-value |
|---|---|---|---|
| N | 427 | 288 | |
| Age (SD), range | 40.21 (11.59), 18–79 | 31.50 (8.92), 18-68 | **<0.001** |
| Gender | | | 0.079 |
| Male | 191 (44.73%) | 123 (42.71%) | |
| Female | 224 (52.46%) | 147 (51.04%) | |
| Other | 12 (2.81%) | 18 (6.25%) | |
| Education | | | **<0.001** |
| High School or equivalent | 45 (10.54%) | 65 (22.57%) | |
| Some College | 88 (20.61%) | 102 (35.41%) | |
| College degree | 218 (51.05%) | 94 (32.64%) | |
| Graduate degree | 76 (17.80%) | 27 (9.38%) | |
| Household Income | | | 0.142 |
| Under $25,000 | 56 (13.11%) | 36 (12.50%) | |
| $25–49,999 | 111 (26.00%) | 65 (22.57%) | |
| $50–74,999 | 89 (20.84%) | 57 (19.79%) | |
| $75–99,999 | 61 (14.29%) | 32 (11.11%) | |
| $100–124,999 | 28 (6.56%) | 27 (9.38%) | |
| $125–149,999 | 21 (4.92%) | 10 (3.47%) | |
| $150–174,999 | 18 (4.21%) | 19 (6.59%) | |
| Over $175,000 | 36 (8.43%) | 30 (10.42%) | |
| Don't Know | 7 (1.64%) | 12 (4.17%) | |
| Race | | | 0.618 |
| White, non-Hispanic | 277 (64.87%) | 184 (63.89%) | |
| Black/African American, non-Hispanic | 31 (7.26%) | 15 (5.21%) | |
| Hispanic | 59 (13.82%) | 46 (15.97%) | |
| Other | 60 (14.05%) | 43 (14.93%) | |
| Psychopathic Traits (TriPM), M(SD) | 55.86 (19.90) | 122.25 (16.92) | **<0.001** |
| Antisocial Behavior (STAB), M(SD) | 58.81 (18.07) | 95.27 (18.67) | **<0.001** |
| Fluid Intelligence, M(SD) | 5.57 (2.09) | 5.16 (1.96) | **0.008** |

p-values were obtained with a two-tailed t-test for continuous variables and a chi-squared test for categorical variables. p < 0.05 in bold.

PCL:SV can be used in forensic and non-forensic settings. Although psychopathy is now agreed to be continuously distributed, scores ≥ 18 have been previously used as clinical cutoffs[73], although scores ≥ 8 optimize specificity and sensitivity for predicting outcomes such as community violence in civil psychiatric samples[75]. The TriPM is moderately correlated with the Psychopathy Checklist Revised[76,77]. However, the PCL-SV captures features of psychopathy not assessed by the TriPM, namely a greater focus on antisociality and criminal behavior. In the current study, PCL-SV was administered via the web conferencing tool Zoom by 2–4 trained interviewers. This 45–60 minute interview asks questions regarding early life experiences, work and relationship history, behaviors, and criminal offenses. Participants who completed the PCL-SV were compensated an additional $20.

## Procedure

All participants completed identical screening and survey measures presented in a randomized order before the social discounting task. In addition to the TriPM[12], antisocial behavior and attitudes were assessed using the Subtypes of Antisocial Behavior questionnaire (STAB)[78]. This is a 32-item self-report measure that assesses rule-breaking behavior, physical aggression, and social aggression in the past year. Fluid intelligence was assessed with a measure drawn from UK Biobank[79], which includes 13 logic and reasoning questions that participants have up to two minutes to answer. Participants also self-reported their lifetime criminal history, including offenses committed, criminal charges, and convictions, along with their incarceration history. Lastly, participants provided demographic information, including age, gender, and household income.

Participants also completed the social discounting task as a part of the online Qualtrics survey (Fig. S1). Following established procedures[46,47,51,52], participants were asked to imagine a list of 100 people, with 1 being their closest other and 100 being a stranger. They were instructed to provide the names of real people they know who represent seven specific social distances ($N$ = 1, 2, 5, 10, 20, 50, 100). Participants then made nine dichotomous choices for each N (Fig. S1). Therefore, there were seven blocks with nine trials (i.e., dichotomous choices) per block. In each trial, the selfish choice entailed choosing to keep an amount of money (the value ranged from $155-$75, decreasing in increments of $10 across trials), while the generous choice remained the same (splitting the money with that N, so both would receive $75). Thus, for example, in one trial, a participant might decide to keep $155 or to split $150 with their closest social other ($N$ = 1). Participants who choose the generous option would thus choose to sacrifice $80 to benefit that person. This format permits an "indifference point" which we estimated for each social other (N) as the trial in which the participant switched from selfish choices to sharing with the other person.

If the participant chose the selfish option for all the trials in a given block, the indifference point was assumed at $75. If the participant chose the generous option for all trials in the block, the indifference point was assumed at $155. Amounts willing to forgo (v) were calculated by subtracting $75 from the indifference point for each block for each participant, resulting in seven "amount willing to forgo" (v) observations corresponding to one of seven social others (N) for each participant (i).

## Statistical analysis

We compared social discounting between groups by calculating the indifference point, which represents the maximum amount participants were willing to forgo for each social distance (N). To determine the best-fitting model for the data, we assessed the Akaike Information Criterion (AIC) values across hyperbolic, exponential, and linear models. The hyperbolic model yielded the lowest AIC value (46142.77), indicating a superior fit compared to the exponential (AIC: 49021.74) and linear (AIC: 48269.92) models. ΔAIC values also confirmed the hyperbolic model was a stronger fit than the exponential model (3,127.15) or linear model (2,127.15) (ΔAIC > 10 indicates strong evidence for the model with the lower AIC value having a superior fit[80]. Finally, model weights (AICcWt) calculated using the *AICcmodavg* package[81] in R indicated that the hyperbolic model had an Akaike weight of 1.00, indicating it was overwhelmingly the most likely model given these data. A hyperbolic discounting curve was thus modeled to estimate discounting rates for each participant. This curve follows the function[46,47]:

$$v = V0/1 + kN \qquad (1)$$

where $V0$ is the value of the reward which stays constant, k is the degree of discounting, N is the social distance, and v is the discounted value of the reward as a function of discounting rate and social distance. Participants' social discounting rates calculated from the hyperbolic model are represented using $k/logk$. Logk represents the rate of decay in the amount a participant is willing to forgo changes as social distance increases.

We conducted secondary analyses using the *pracma* package[82] in R to calculate the area under the curve (AUC), which is a model-agnostic assessment of discounting. AUC is calculated using a trapezoidal function that sums the average amount a participant is willing to forgo for each social distance and converts this value to a proportion between 0 and 1, with higher AUC scores reflecting more overall generosity. AUC and logk thus provide

unique insights, as AUC captures overall generosity and logk highlights the rate of change. All models also included age, gender (with female/other as the reference), household income, and fluid intelligence as covariates. Results regarding AUC values are found in the Supplementary Information file (Tables S18–26) and are similar to results using logk.

Multiple regression models were used to examine the relationship between psychopathy and social discounting (dichotomously and continuously), and whether social discounting moderated group differences in antisocial behavior. Mediation analyses were completed using the *mediation* package[83] in R to investigate if social discounting explained group differences in antisocial behavior. Lastly, multiple regression models were used to investigate if age moderated the relationship between psychopathy and social discounting. All analyses were conducted using two-tailed tests. All models were checked for standard statistical assumptions, and assumptions were generally met across models. To examine the robustness of our primary results, we conducted two robustness checks using 10-fold cross-validation and running models following propensity score matching. Results of these analyses were consistent with those presented below and are reported in the Supplementary Information file (*cross-validation*: Table S2; *propensity score matching*: Tables S3, S4).

### Reporting summary
Further information on research design is available in the Nature Portfolio Reporting Summary linked to this article.

## Results
Summed total psychopathy scores for each participant were calculated along with subscale scores. The mean total psychopathy score for participants in the high-psychopathy group was 122.25 (SD = 16.92, overall range = 91-165; Female/Other = 91-163, Male = 105-165). This places all high psychopathy participants above the 95th percentile of TriPM scores according to their gender, with their mean score being above the 99th percentile[4]. Controls' mean score was 55.86 (SD = 19.90, overall range = 19–104, Female/Other = 19–90, Male = 23–104; $t(675.83) = −47.90$, $p < 0.001$, Cohen's d = −3.78, 95% CI [−3.78, −3.30]), placing controls between the 1st and 94th percentile with mean score being at the 39th percentile for females and 28th percentile for males[4] (Table S5). Groups did not differ in gender composition or race/ethnicity (Table 1). Because only age ranges can be pre-specified in CloudResearch the control group was older, $t(700.63) = 11.33$, $p < 0.001$, d = 0.82, 95% CI [0.66, 0.98], Table 1, as well as more educated, $t(590.42) = 6.90$, $p < 0.001$, d = 0.53, 95% CI [0.38, 0.68], Table 1, and higher in fluid intelligence, $t(643.03) = 2.66$, $p = 0.01$, d = 0.20, 95% CI [0.05, 0.35], Table S5, than the high psychopathy group.

Given group differences and prior evidence linking these variables to prosociality and antisociality[50,67,84,85], age and fluid intelligence were included as covariates in all models. Income and gender were also included as covariates, given consistent evidence linking them to prosociality[50,62,78]. (19 participants who reported not knowing their household income were re-coded as being in the mean income bracket of the full sample.) Group differences in TriPM scores persisted when controlling for age, gender, income, and fluid intelligence ($B = 0.83$, $p < 0.001$, 95% CI [0.79, 0.87], $F(5,709) = 473.9$). Across the full sample, high internal consistency of total psychopathy scores (Cronbach's a α = 0.97), Boldness (α = 0.90), Meanness (α = 0.96), and Disinhibition (α = 0.94) subscale scores was observed. To further assess the reliability of responses, we conducted Cronbach's alpha reliability analyses for the TriPM total score and subscales with each recruitment source separately. We also compared responses across groups to the pre-survey commitment request (see Supplementary Information). Results indicated high reliability of responses for both groups and comparable responses to the commitment request.

Forty-four participants in the high psychopathy group completed a PCL:SV interview. Ratings were determined from the semi-structured interview. All interviewers received formal training in administering and scoring PCL instruments. Following each interview, each interviewer independently scored the participant before the raters met to discuss and

decide on final scores for each of the 12 items. The average PCL:SV final score was 14.32 (SD = 5.18), with scores ranging from 3–23. 38/44 (86%) of screened participants received scores >= 8. Total PCL:SV and TriPM scores were moderately correlated, $r(42) = 0.49$, $p < 0.001$, 95% CI [0.23, 0.69], and within the range found in prior work ($rs = 0.20-0.62$)[76,86]. Interrater reliability of Total, Factor 1, and Factor 2 scores was estimated using intraclass correlation coefficients (ICC) using the *psych* R package[87] based on a single rater, absolute, one-way random-effects model. ICCs were 0.81 ($p < 0.001$, 95% CI [0.73, 0.88]) for total scores, 0.72 ($p < 0.001$, 95% CI [0.62, 0.82]) for Factor 1, and 0.76 ($p < 0.001$, 95% CI [0.67, 0.85]) for Factor 2[88].

We also found high internal consistency for total antisociality (STAB) scores across the sample (α = 0.96) and within each group (see Supplementary Information) and thus calculated each participant's summed antisocial behavior. Consistent with their psychopathy scores, high-psychopathy participants reported significantly more antisocial behavior ($M = 95.27$, $SD = 18.67$, range = 44–155) than controls ($M = 58.81$, $SD = 18.07$, range = 32–138), $t(602.27) = −25.94$, $p < 0.001$, d = −1.99, 95% CI [−2.17, −1.81] (Table 1, Table S5). Group differences persisted when controlling for age, gender, income, and fluid intelligence, $B = 0.68$, $p < 0.001$, 95% CI [0.63, 0.74], (Table S6). High-psychopathy participants also reported committing, being charged with, and being convicted of criminal offenses at much higher rates than controls. High-psychopathy participants were 1199% more likely to have committed at least one crime, 287% more likely to have been charged with at least one crime, and 209% more likely to have been convicted of at least one crime (again, controlling for covariates; committed: OR = 12.99, 95% CI [8.48, 20.39], $p <0.001$; charged: OR = 3.87, 95% CI [2.61, 5.80], $p < 0.001$; convicted: OR = 3.09, 95% CI [2.06, 4.67], $p < 0.001$; Tables S7-S9). Associations between group and criminal involvement remained statistically significant following Bonferroni correction to α = 0.017 across the three group comparisons. The most frequently reported criminal behaviors committed by respondents in the high-psychopathy group were drug possession (64%), driving under the influence (61%), reckless driving (59%), vandalism (56%), larceny (53%), and assault (42%) (Table 2).

After modeling the group differences in the social discounting curve using a hyperbolic model (Table S10), individual logk values were calculated for each participant. A bivariate association between psychopathy group and logk was observed, $t(614.49) = −11.97$, $p < 0.001$, d = −0.91, 95% CI [−1.07, −0.76], which persisted after controlling for age, gender, income, and fluid intelligence, $B = 0.38$, $p < 0.001$, 95% CI [0.31, 0.45]; Table 3, Fig. 1), indicating that high-psychopathy participants show a significantly steeper hyperbolic decay in generosity as social distance increases relative to controls. Results were replicated when utilizing 10-fold cross-validation and propensity score matching, reported in the Supplementary Information file (*cross-validation*: Table S2; *propensity score matching*: Tables S3–4). No statistically significant main effects of gender or income were observed. However, a main effect of age was observed, such that social discounting decreased as age increased, $B = −0.13$, $p = 0.001$, 95% CI [−0.20, −0.05], and a main effect of fluid intelligence was observed, such that social discounting increased as intelligence increased, $B = 0.10$, $p = 0.005$, 95% CI [0.03, 0.16]. Similar results were observed when the relationship between discounting (logk) and psychopathy was assessed as a continuous measure across all participants, with social discounting again increasing as psychopathy increased, $B = 0.40$, $p < .001$, 95% CI [0.33, 0.47] (Table 4, Fig. S2), with the bivariate association also being significant, $r(713) = 0.43$, $p < 0.001$, 95% CI [0.36, 0.48]. Associations between psychopathy and logk remained statistically significant following Bonferroni-correction to α = 0.025 to account for conducting both dichotomous and continuous tests.

To identify whether one or more subscales of psychopathy were driving this association between psychopathy and logk, we conducted a multiple linear regression including meanness, disinhibition, and boldness subscale scores as predictors of logk, with age, gender, income, and fluid intelligence included as covariates. Meanness was the only subscale that predicted logk, $B = .41$, $p < 0.001$, 95% CI [0.29, 0.54], (Table S11). The bivariate correlation between meanness and logk was $r = 0.45$ ($df = 713$, $p < 0.001$). Results were

**Table 2 | Criminal history in order of prevalence of crimes committed in the high psychopathy group**

| | Control (N = 427) | | | High Psychopathy (N = 288) | | |
|---|---|---|---|---|---|---|
| | Committed | Charged | Convicted | Committed | Charged | Convicted |
| Any | 178 (41.69%) | 86 (20.14%) | 73 (17.10%) | 248 (86.11%) | 110 (38.19%) | 92 (31.94%) |
| Drug Possession | 85 (19.91%) | 18 (4.22%) | 19 (4.50%) | 184 (63.89%) | 45 (15.63%) | 34 (11.81%) |
| DUI | 82 (19.20%) | 28 (6.56%) | 29 (6.79%) | 176 (61.11%) | 29 (10.07%) | 27 (9.38%) |
| Reckless Driving | 67 (15.69%) | 19 (4.45%) | 18 (4.22%) | 170 (59.03%) | 32 (11.11%) | 23 (7.99%) |
| Vandalism | 45 (10.54%) | 3 (0.70%) | 1 (0.23%) | 161 (55.90%) | 12 (4.17%) | 12 (4.17%) |
| Larceny | 52 (12.18%) | 12 (2.81%) | 11 (2.58%) | 154 (53.47%) | 25 (8.68%) | 16 (5.56%) |
| Assault | 27 (6.32%) | 15 (3.51%) | 8 (1.87%) | 122 (42.36%) | 33 (11.46%) | 20 (6.94%) |
| Intent to sell drugs | 32 (7.49%) | 5 (1.17%) | 2 (0.47%) | 113 (39.24%) | 14 (4.86%) | 12 (4.17%) |
| Truancy | 38 (8.90%) | 4 (0.94) | 1 (0.23%) | 107 (37.15%) | 15 (5.21%) | 13 (4.51%) |
| Weapon Possession | 14 (3.28%) | 4 (0.94%) | 4 (0.94%) | 94 (32.64%) | 13 (4.51%) | 11 (3.82%) |
| Running Away | 24 (5.62%) | 5 (1.17%) | 4 (0.94) | 88 (30.56%) | 18 (6.25%) | 15 (5.21%) |
| Burglary | 16 (3.75%) | 3 (0.70%) | 4 (0.94%) | 82 (28.47%) | 11 (3.82%) | 7 (2.43%) |
| Arson | 5 (1.17%) | 1 (0.23%) | 1 (0.23%) | 58 (20.14%) | 5 (1.74%) | 2 (0.69%) |
| Robbery | 8 (1.87%) | 6 (1.41%) | 5 (1.17%) | 56 (19.44%) | 9 (3.13%) | 7 (2.43%) |
| Prostitution | 4 (0.94%) | 0 | 2 (0.47%) | 51 (17.71%) | 2 (0.69%) | 2 (0.69%) |
| Auto Theft | 9 (2.11%) | 5 (1.17%) | 5 (1.17%) | 27 (9.38%) | 5 (1.74%) | 6 (2.08%) |
| Rape | 2 (0.47%) | 4 (0.94) | 5 (1.17%) | 14 (4.86%) | 1 (0.35%) | 2 (0.69%) |
| Murder | 4 (0.94%) | 1 (0.23%) | 3 (0.70%) | 7 (2.43%) | 5 (1.74%) | 3 (1.04%) |
| **Gun Violence** | | | | | | |
| Shoot | 6 (1.41%) | | | 17 (5.90%) | | |
| Robbery | 3 (0.70%) | | | 16 (5.56%) | | |
| Gang | 5 (1.17%) | | | 14 (4.86%) | | |
| Kill | 4 (0.94) | | | 10 (3.47%) | | |
| Carjack | 6 (1.41%) | | | 3 (1.04%) | | |

**Table 3 | Psychopathy group predicting logk**

| Variable | b (se) | CI | Std. B (se) | Std. CI | p |
|---|---|---|---|---|---|
| (Intercept) | −2.20 (0.35) | −2.89−−1.51 | 0.00 (0.03) | −0.07–0.07 | **<0.001** |
| High Psychopathy > Controls | 1.56 (0.15) | 1.26–1.86 | 0.38 (0.04) | 0.31–0.45 | **<0.001** |
| Age | −0.02 (0.01) | −0.03−−0.01 | −0.12 (0.04) | −0.19−−0.05 | **0.001** |
| Gender (Male > Female/Other) | 0.12 (0.14) | −0.14–0.39 | 0.03 (0.03) | −0.04–0.10 | 0.37 |
| Income | −0.07 (0.03) | −0.13−−0.001 | -0.07 (0.03) | −0.14−−0.001 | 0.05 |
| Fluid Intelligence | 0.10 (0.03) | 0.03–0.16 | 0.10 (0.03) | 0.03–0.16 | **0.005** |
| | $F(5,709) = 34.62$, $p <.001$, Adjusted $R^2 = 0.19$ | | | | |

p < 0.05 in bold

replicated when utilizing 10-fold cross-validation and propensity score matching, reported in the Supplementary Information file (*cross-validation:* Table S2; *propensity score matching:* Tables S3, 4).

**Do differences in social discounting mediate differences in antisocial behavior?**
We also found a bivariate association between logk and antisocial behavior across the full sample ($r(713) = 0.35$, $p < 0.001$, 95% CI [0.29, 0.42]), which remained statistically significant after controlling for age, gender, income, and fluid intelligence, $B = 0.30$, $p < 0.001$, 95% CI [0.23, 0.37], $F(5,709) = 33.38$. Mediation analysis found that social discounting (logk) partially mediated group differences in antisocial behavior (total effect: $p < 0.001$, direct effect: $p < 0.001$, indirect effect: $p = 0.018$; proportion mediated: $b = 0.04$, $p = 0.018$, 95% CI [0.006, 0.08]; Table S12]. This indicated that increased antisocial behavior in the high psychopathy group is partly explained by increased discounting rates. The mediating effect was not significant when considering psychopathy as a continuous variable across the full sample (total effect: $p < 0.001$, direct effect: $p < 0.001$, indirect effect: $p = 0.67$; proportion mediated: $b = 0.006$, $p = 0.67$, 95% CI [−0.02, 0.03]; Table S13). Associations between psychopathy and logk remained statistically significant following Bonferroni-correction to α = 0.025 to account for conducting both dichotomous and continuous tests of mediation.

We also observed a significant interaction between discounting and group in predicting antisocial behavior, $B = 0.09$, $p = 0.002$, 95% CI [0.03, 0.15] (Table S14; Fig. 2), such that as social discounting (logk) increases, antisocial behavior increases at a higher rate in the high-psychopathy group relative to controls. However, there was not a significant interaction when examining whether social discounting moderated the relationship between psychopathic traits (measured continuously) and antisocial behavior across the full sample, $B = 0.0$, $p = 0.85$, 95% CI [−0.04, 0.05] (Table S15).

**Fig. 1 | Hyperbolic social discounting curve across studies.** Social discounting curves with standard error for the current study (control and high psychopathy group, $N = 715$) and Vekaria et al (2017) (controls, $N = 26$). The high psychopathy group had higher discounting than both control groups, with control groups overlapping. The shaded region represents the standard error around the mean.

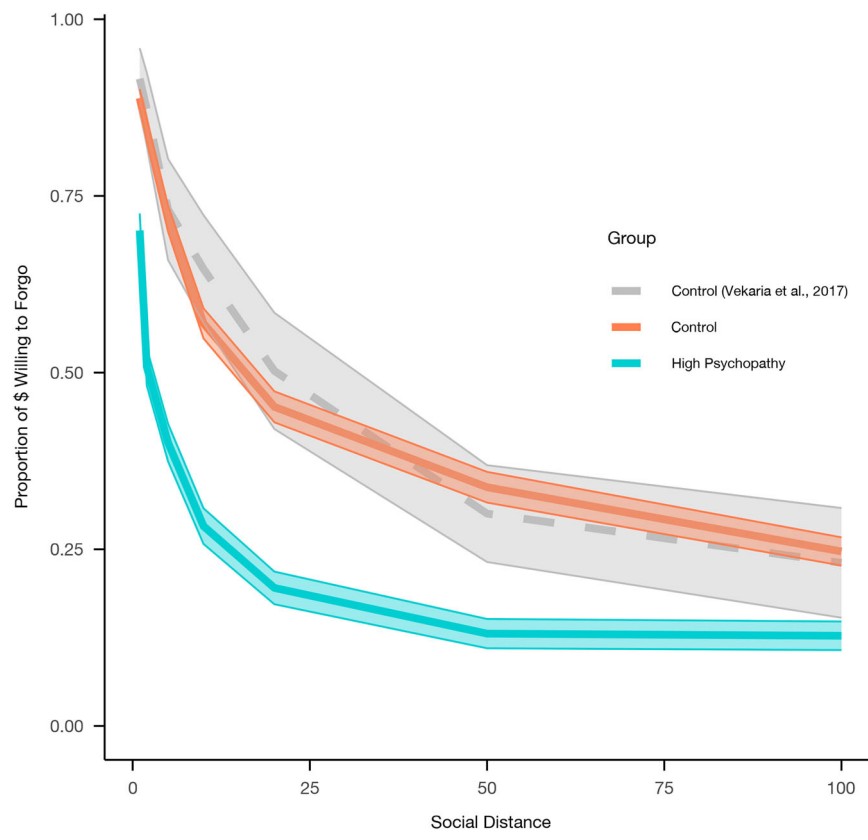

## Table 4 | Psychopathic traits predicting logk

| Variable | b (se) | CI | Std. B (se) | Std. CI | p |
|---|---|---|---|---|---|
| (Intercept) | −3.41 (0.41) | −4.21−−2.61 | 0.00 (0.03) | −0.07−0.07 | **<0.001** |
| Psychopathic Traits (TriPM) | 0.02 (0.00) | 0.02−0.03 | 0.40 (0.04) | 0.33−0.47 | **<0.001** |
| Age | −0.02 (0.01) | −0.03−−0.01 | −0.10 (0.04) | −0.18−−0.03 | **0.005** |
| Gender (Male > Female/Other) | −0.04 (0.14) | −0.31−0.23 | −0.01 (0.03) | −0.08−0.06 | 0.76 |
| Income | −0.07 (0.03) | -0.14−−0.01 | −0.08 (0.03) | −0.14−−0.01 | **0.03** |
| Fluid Intelligence | 0.10 (0.03) | 0.03−0.17 | 0.10 (0.03) | 0.03−0.17 | **0.003** |
| | $F(5,709) = 36.86$, $p < .001$, Adjusted $R^2 = 0.20$ | | | | |

$p < 0.05$ in bold

Bonferroni-correction was applied ($\alpha = 0.025$), and the group-based interactions remained significant under this threshold.

### Does age moderate the relationship between psychopathic traits and social discounting?

Prosociality has been observed to increase with increasing age[67] and antisocial behavior in psychopathy declines with age[1,89,90]. In addition to observing a negative relationship between age and social discounting, we also replicated the negative association between antisocial behavior and age across the full sample ($B = -0.31$, $p < 0.001$, 95% CI [−0.38, −0.24], $F(4,710) = 21.72$). However, when examining if age moderated group differences, the high psychopathy group exhibited a positive association between age and antisocial behavior, whereas the opposite association was observed in controls when controlling for gender, income, and fluid intelligence, Age x Group $B = 0.12$, $p < 0.001$, 95% CI [0.06, 0.18], $F(6,708) = 123.1$. We therefore conducted a multiple regression analysis to determine whether age moderates the relationship between psychopathic traits and social discounting. We did not find evidence that age moderated the relationship between psychopathy (treated either dichotomously or continuously) and discounting indexed by logk, dichotomous: $B = 0.04$,

$p = 0.31$, 95% CI [−0.04, 0.12], (Table S16); continuous: $B = 0.03$, $p = .39$, 95% CI [−0.04, 0.10], (Table S17). Thus, although age and antisocial behavior are positively correlated in the high psychopathy group, and the reverse association was observed in controls, age did not significantly moderate the relationship between psychopathy and discounting. However, moderation models involving age showed slight evidence of heteroscedasticity, so findings should be interpreted with caution.

## Discussion

We report the results of a study involving a large community-recruited sample of very-high psychopathy adults and demographically similar controls, which found that psychopathy was associated with robust, hyperbolic increases in social discounting (logk) and reduced overall generosity (AUC). These associations were mainly driven by the Meanness subscale scores, and social discounting partially mediated group differences in antisocial behavior. These findings indicate that adults with high levels of psychopathy subjectively devalue the welfare of others relative to controls, and link social discounting to increased antisocial behavior in this population.

Our sample is among the largest reported studies of adults in the community with very high psychopathy scores ($n = 288$), with mean

**Fig. 2 | logk predicting antisocial behavior in high psychopathy and controls.** This figure shows predicted antisocial behavior from the regression model with 95% confidence interval. In the high psychopathy group, antisocial behavior increases with increased discounting, but there is no association in controls ($N = 715$).

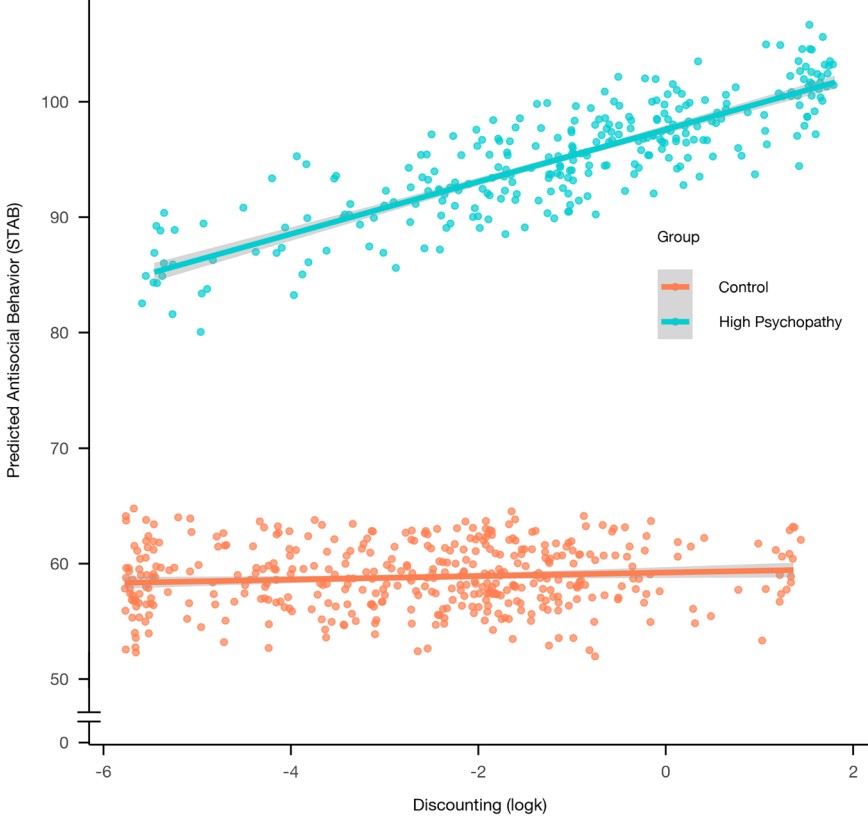

psychopathy scores above the top 1% for their gender[4]. These participants reported significantly more antisocial behavior than typical adults, and our results indicate a robust relationship between psychopathy and antisocial behavior even among these community-recruited participants. Building on previous findings of reduced prosociality in psychopathy[40–43], our findings extend this work by demonstrating that people high in psychopathy—particularly those scoring high on Meanness—place lower subjective value on others' welfare, and this association statistically accounts for their increased antisocial behavior.

These results also potentially speak to the question of whether antisocial behaviors, including criminal behaviors, are an intrinsic feature versus a downstream correlate of psychopathy. Psychopathy is a personality construct that reflects several sub-components that vary continuously across the population[4,11] and that include traits like meanness and narcissism that indicate devaluation of others' welfare and are robust predictors of antisocial behavior[4,91,92]. Our results suggest that antisocial behaviors that reduce others' welfare may be intrinsically potentiated by very low subjective valuation of others' welfare in high-psychopathy adults, even when psychopathy is assessed using triarchic measures like the TriPM that de-emphasize criminal and antisocial behavior relative to PCL-based assessments[12,93]. This may reflect the close association between devaluation of others' welfare and the meanness subscale, which is a core feature of all major psychopathy measures.

Our recruitment approach enabled both continuous and group-based analyses using a multiverse framework. The convergence of results across continuous and group-based analyses, as well as across two different indices of social discounting (logk and AUC), supports the robustness of our findings. Results indicated some non-linear effects across our sample. Social discounting moderated the relationship between psychopathy group and antisocial behavior, predicting antisocial behavior in the high psychopathy group to a greater degree than in the control group. We also found that antisocial behavior decreased with age in typical adults but increased with age in high-psychopathy adults. These findings support prior work indicating that the association between psychopathy and relevant outcome

variables is not always linear[20,94] such that typical community samples may not reveal patterns observed in high-psychopathy samples.

Our recruitment approach yielded a sample diverse in gender, age, and other variables, unlike many institutional samples or undergraduate samples[95], enabling us to test effects that are difficult to detect in more homogeneous or lower-variance samples. We could thus consider effects of income and gender in our analyses; however, we did not observe significant associations despite prior research linking these variables to generosity[96–99]. This is consistent with prior research[52], which has suggested that the social discounting task is not simply a donation task, but more generally indexes the subjective valuation of others' outcomes relative to one's own outcomes. We did find that social discounting declined with age, consistent with prior findings[67], but we did not find statistical evidence that age moderated the relationship between psychopathy and social discounting.

These findings contrast with prior studies of psychopathy and social discounting, which have found small or null effects ($rs = 0$ to $-0.19$)[40,100]. Features of our sample and task may partly explain the divergent findings. Two prior studies of social discounting in psychopathy in undergraduates did not assess choices about targets who varied linearly in social distance as the standard task does, but about people categorized as close, neither close nor distant, distant, and very distant, finding null or small effects (rs = $-0.06$ to .19)[53,54]. However, by using specific increments of social distance, the standard task enables precise estimation of discounting curves and more direct interpretation of subjective value. Our larger effect sizes (total psychopathy: $r = 0.35$; meanness: $r = 0.40$) are therefore unlikely to be the result of chance fluctuation due to our larger sample size. They more likely reflect our wider range of psychopathy scores, including very high scores, and a standard task modeled on prior social and temporal discounting paradigms, allowing us to interpret our findings as indicating decreased subjective valuation of others' welfare as psychopathy increases.

Our findings have implications for psychopathy research conducted in student samples, as undergraduate participants score lower on psychopathy dimensions and exhibit restricted variance. As a result, effects observed in clinical samples—in this case, the strong relationship between social

discounting and antisocial behavior—may be attenuated in student samples[101]. More explicit efforts by researchers to identify similarities and differences between observed patterns in clinical and subclinical psychopathy may be valuable.

## Limitations

This study's results should be interpreted in the context of some limitations. First, like many clinical research studies and other studies recruiting special populations[102], this study employed a cross-sectional, correlational design using a purposive sample. We thus refrain from drawing causal conclusions about the origin of the observed effects. Because most of our high-psychopathy participants were recruited after seeking information about psychopathy online, they were in part, self-selected. Self-selection is a pervasive consideration in psychological research, as research participation is voluntary even in institutional and clinical settings, such that research participants represent a small fraction of potentially eligible adults and may generally be biased toward populations that are disproportionately female, wealthy, and/or prosocial[103–105]. A meta-analysis of psychopathy as assessed by the TriPM[101] observed that the specific sample studied (e.g., undergraduates, imprisoned samples) moderates the nature of the observed associations. Lending some support to the generalizability of our results, we observed similar results whether including or excluding high-psychopathy CloudResearch panel participants. We also observed high and consistent self-reported commitment and response reliability across participant groups. However, our sample may nonetheless be non-representative of high-psychopathy adults in the community. We evaluated the potential role of covariates such as age, gender, household income, and fluid intelligence in social discounting, but other potentially relevant factors not measured here could include housing status, early-life adversity, or childhood household income. Lastly, when completing the social discounting task, our participants allocated hypothetical resources rather than real money. This approach is supported by previous studies[46,51], but may affect choice patterns[106]. Although some studies suggest that using actual rewards may reduce discounting magnitude[107], the results of a recent meta-analysis indicate that the use of hypothetical rewards does not affect discounting patterns[71], and we are aware of no evidence that this would affect higher psychopathy participants differently.

## Conclusion

The antisocial behaviors associated with psychopathy (including financial, legal, and medical expenses) are estimated to yield societal costs of over $460 billion annually[108]. It is thus crucial to understand factors influencing antisocial behavior in high-psychopathy populations. This research indicates that psychopathy may be characterized by placing a very low subjective value on others' welfare, increasing high-psychopathy individuals' risk for engaging in behaviors that are harmful to others. Identifying the origins of this feature of psychopathy and treatments that may ameliorate it are important goals for future research.

## Data availability

Data and materials, including raw data and processed data, are publicly available[70] at https://doi.org/10.17605/OSF.IO/8DF4N.

## Code availability

Analysis was completed using R version 4.4.1. Analysis code can be publicly accessed[70] at https://doi.org/10.17605/OSF.IO/8DF4N.

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

## Acknowledgements

We are grateful to the participants who contributed their time and energy to this work. A.A.M. discloses support for the research of this work from the Lisa Heidi Michael Gift Fund and the National Science Foundation (NSF; grant # 2139925). N.N. discloses support for the research of this work from the Georgetown Patrick Healy Fellowship. The funders had no role in study design, data collection and analysis, decision to publish, or preparation of the manuscript.

## Author contributions

Conceptualization by N.N., M.D., P.F., and A.A.M.; Methodology by N.N., M.D., P.A., P.F., and A.A.M.; Formal Analysis by N.N., P.A., M.D., and A.A.M.; Investigation by N.N., M.D., P.F., M.S., L.P., K.D., V.A.T, and A.A.M.; Writing – original draft by N.N. and A.A.M.; Writing – review and editing by N.N., M.D., P.A., P.F., M.S., L.P., K.D., V.A.T., and A.A.M.; Funding Acquisition by N.N. and A.A.M.

## Competing interests

The authors declare no competing interests.
