## [Transparent Peer Review file · Communications Psychology]

Adults with More Severe Psychopathy in the Community Show Increased Social Discounting

Corresponding Author: Ms Naomi Nero

Version 0:

Decision Letter:

Dear Ms Nero,

Thank you for your patience during the peer-review process. Your manuscript titled "Severity of Psychopathy in a Community-Recruited Sample is Indexed by Increased Social Discounting" has now been seen by 3 reviewers, and I include their comments at the end of this message. They find your work of interest but raised some important points. We are interested in the possibility of publishing your study in Communications Psychology, but would like to consider your responses to these concerns and assess a revised manuscript before we make a final decision on publication.

We therefore invite you to revise and resubmit your manuscript, along with a point-by-point response to the reviewers. Please highlight all changes in the manuscript text file.

Editorially, we consider it crucial that any specified hypotheses a priori are clearly stated in the introduction in the revised manuscript. In addition, please thoroughly address the reviewers concerns regarding the potential causal claim with the observational design, as well as other methodological concerns such as the statistical approaches used and the choice of covariates.

Please ensure you follow our statistical guidelines when reporting statistics (<https://www.nature.com/commpsychol/submit/submission-guidelines#statistical-guidelines>). Please note in particular our requirements for the reporting and interpretation of null-results. Non-significant findings derived from null-hypotheses significance tests should be reported in full, but may not be interpreted. Where you interpret null results, this interpretation must be based on Bayes Factors or equivalence tests.

I am attaching an Editorial Requests Table that details critical reporting requirements for the revised manuscript. Please attend to each item and ensure your manuscript is fully compliant. If your revised manuscript is not aligned with these requests on major issues, such as those concerning statistics, it may be returned to you for further revisions without re-review.

Please submit the following items:

- Revised manuscript
- Point-by-point response to the referees' comments
- Cover letter (as a separate document)
- <https://www.nature.com/documents/nr-reporting-summary.pdf>>Nature Research Reporting Summary

- Completed Editorial Request Table (attached).

via this link: Link Redacted .

Additional guidance is available in our style and formatting guide Communications Psychology formatting guide.

Best regards,

Troby Lui, on behalf of

Inti Brazil

Troby Lui, PhD
Associate Editor
Communications Psychology

Inti Brazil, PhD
Editorial Board Member
Communications Psychology
orcid.org/0000-0001-5824-0902

REVIEWER EXPERTISE:

Reviewer #1: psychopathic traits, social decision making, neuroscientific methods

Reviewer #2: psychopathy, cognitive functions, neuroimaging

Reviewer #3: psychopathy, social cognition, methods

REVIEWER REPORTS:

Reviewer #1 (Remarks to the Author):

Thank you for the opportunity to review this manuscript, which investigates the relationship between psychopathic traits and social discounting in a large, community-recruited sample. This is a timely and important contribution, especially given the need for research on non-institutionalized individuals with high psychopathy. The manuscript is generally well-written, the analyses are clearly presented, and the study is methodologically ambitious.

That said, I offer several suggestions to enhance the clarity, contextual grounding, and methodological rigor of the paper. These are detailed below.

Citations and Framing the Contribution

The authors could enhance the impact and contextual clarity of their manuscript by expanding their citation practices to better reflect recent research and include a broader range of voices. A significant proportion of the cited literature—approximately 60%—is over a decade old. While many of these are foundational papers that rightly inform the theoretical foundation, the field has evolved substantially with more nuanced findings from work that has been testing ideas from these > 10 year old ideas that are highly relevant to the present work.

For instance, recent contributions by Nikolaus Steinbeis, who explores cognitive and neural mechanisms of prosocial and antisocial behavior, and Joseph Sakai, who has developed paradigms measuring prosocial behavior toward distant others in relation to psychopathic traits, offer frameworks that could enrich the current manuscript's discussion. Including these perspectives would not only align the manuscript with the current state of the field but would also provide readers with a

more comprehensive understanding of how this study advances existing knowledge.

This adjustment would support a more inclusive and up-to-date discussion and offer the opportunity to situate the findings within current debates in the literature on psychopathy and prosocial cognition.

Unpublished Citation

The manuscript cites an "in prep" manuscript:

Berluti, K., Ploe, M., Doherty, H., Jones, D., Patrick, C., Marsh, A.A. (2024). Prevalence and correlates of psychopathy in the general population. In prep.

Because there is no accessible preprint or publication associated with this reference, it is not possible to verify or evaluate the cited findings. Given that "in preparation" manuscripts are inherently unstable and may change substantially prior to publication, I recommend that this citation either be removed or replaced with a citable preprint if available.

Study Design and Methodology

The study employs an observational design using a convenience sample without random assignment. While this approach may be appropriate for the population of interest, it does necessitate a cautious interpretation of causal claims. I strongly encourage the authors to include a more substantial discussion in the limitations section addressing how the lack of randomization and self-selection may impact internal validity. I note the authors tried to get ahead of review comments by talking about how self-selection is an issue but I do not feel what is provided in this context is sufficient – as this is the crux for reliance on the outcomes of this work and is a very probable area of bias.

Key point: The paper acknowledges demographic differences between groups (age, education, fluid intelligence), which were controlled for in statistical models. However, the observed differences in core covariates associated both with group membership and outcomes (e.g., age and cognitive ability) raise the potential for collider bias, which can compromise the validity of group comparisons.

To strengthen inference, I suggest the authors consider:

- Propensity score matching to reduce imbalance across covariates
- Sensitivity or robustness checks using permutation testing to assess significance under exchangeability
- Cross-validation procedures to assess the stability of group-level effects

Such steps would provide more robust support for the reported associations and increase confidence in the findings.

Ultimately, these issues will impact confidence in the results regardless and I believe is an important reason to scale back the language. As of now it sounds quite causal and undermines the substantial design issues of the work – please consider substantially revising the language of this article and the title to be more commensurate with the quality of evidence indicative of the design.

Covariate Selection and Control Strategy

The manuscript includes demographic covariates (age, gender, household income, and fluid intelligence) in all models, which is a good practice. However, a more theory-driven approach to selecting covariates could be valuable. Have the authors considered including additional covariates based on theoretical relevance (e.g., childhood adversity, SES beyond income, other confounding mental health symptoms)? Alternatively, including a statement clarifying the rationale for the chosen set would be helpful. Acknowledging the limitations of the current control strategy would also strengthen the discussion.

Sample Validity and Data Quality

The study combines data from two distinct sources: one group recruited through CloudResearch and another via a non-profit website offering psychopathy resources. While the use of community samples is a strength, these recruitment methods differ significantly. CloudResearch applies quality controls for online respondents, while the website group is self-selected and recruited via a psychopathy-specific context.

It would benefit the manuscript to include additional discussion (or ideally analyses) addressing:

- The potential for sampling bias in the self-recruited group
- Whether participant characteristics (e.g., response patterns, reliability) differ by recruitment method
- Any efforts to ensure response validity across groups (e.g., attention checks, repeated measures, or validation items)

The paper notes exclusion of participants failing multiple attention checks, which is reassuring. However, it would be helpful to report any data quality comparisons across recruitment pools and acknowledge the limitations of online, unsupervised testing environments, particularly for a psychopathy sample.

The authors should also consider identifying any patterned responses in the data that may indicate a less reliable way or reporting that is not in good faith and control for this. These are online samples and anything the authors can do to make readers more confident the data is handled appropriately would improve the impact of this work.

Transparency and Code Sharing

The authors commendably share both data and code via the OSF. However, the shared R Markdown script could be improved in clarity. Currently, the code includes dense inline commenting within code blocks, making it difficult to distinguish between executable code and explanatory text.

Best practice suggests:

- Housing detailed explanations and context ****outside**** code chunks
- Using inline comments sparingly within code (e.g., to annotate logic)
- Providing a narrative flow in markdown cells for clarity
- Provide not just the code but also the markdown HTML once it is knitted so others can see the exact output you did when conducting the analysis (as well as data and modeling assumption checks that are not reported on **see below comment**). This would improve reproducibility and accessibility for external readers or future collaborators.

Methods and Diagnostics

The manuscript clearly states that the study was not preregistered. Given the flexibility in analytic choices, a brief post hoc discussion of analytic decision points, degrees of freedom, and how multiple comparisons were handled would be helpful.

It would also strengthen the manuscript if the authors elaborated on why their chosen analytic approach was preferred over more flexible modeling frameworks that could simultaneously account for both group-level and continuous variation. For instance, if I understand the analytic plan correctly, it appears the authors conducted separate analyses to address group comparisons and continuous effects, when in fact a single hierarchical or mixed-effects model could have captured both, while also improving parameter reliability by leveraging the full structure of the data.

Additionally, there is no mention of model diagnostics or checks for statistical assumptions (e.g., residual normality, linearity, multicollinearity). Including a brief note that assumptions were tested that is demonstrated in the code—or acknowledging this as a limitation and providing justification for continuing anyway—would be advisable.

Reviewer #2 (Remarks to the Author):

Thank you for the opportunity to review this manuscript, which I very much enjoyed reading. Indeed, it is very well written and organised. It presents novel findings on a very large sample of individuals spanning low to very high psychopathic scores performing a social discounting task. I do not have any concerns about the methodology, but offer some suggestions, which could help to improve this already excellent manuscript.

Introduction

I was surprised that there were no hypotheses formulated in the last paragraph of the introduction (although the method section states that the hypotheses were not pre-registered, which is a bit confusing), as there's been previous work that is cited by the authors and clearly one would expect a positive association between psychopathic traits and social discounting.

Admittedly, it might be a bit late now to include hypotheses, but I leave the authors and the editor decide on that aspect.

Relatedly, there are analyses examining age effects and moderation by age, but there is no mention in the discussion of age being a potentially important factor that would be formally tested in the manuscript; I think this should be touched on in the introduction.

Methods

For the description of the power analysis, the statistical threshold should be stated (i.e., at $p < .05$ I assume?)

For the PCL-SV description, it would be helpful to state what cut-off is considered indicative of psychopathy, which I believe is 18.

Results

The supplementary material includes up to 44 Supplementary tables(!), but not all of those are mentioned in the main manuscript. Are all those tables really necessary, as they are not mentioned in the main ms?

Also, it would be easier to read the results if all the stats were included in the table in an additional column or 2; that would make the main text of the results lighter.

p.13 This subtitle section: "Do Differences in Social Discounting Correspond to Differences in Antisocial Behavior?" the word "Corresponds" should be replaced by "Mediate" or "Account".

p.14 Mediation analysis: Please state what percentage of psychopathy-ASB association was accounted by the mediator.

Figure 1: Specify what the shaded area represents in the graph.

Reviewer #3 (Remarks to the Author):

Thank you for the opportunity to review the manuscript titled "Severity of Psychopathy in a Community-Recruited Sample is Indexed by Increased Social Discounting". Overall, the topic is highly relevant, and the manuscript is generally well-written and clearly structured. I believe it has the potential to be published in Nature Communications Psychology. That said, I have a few comments and suggestions that may help further improve the manuscript.

Abstract

1) The abstract states that the authors used a "sample of very-high psychopathy adults" (p. 2, line 32). It would be helpful to define what constitutes "very-high" psychopathy, as was done later in the manuscript on page 11 (i.e., "top 95th percentile of TriPM scores").

Introduction

2) My main concern with the introduction is the absence of a review of findings from studies examining psychopathy using economic games and/or the Social Value Orientation (SVO) Slider. Although these paradigms differ from the social discounting task (as the authors note), it would nonetheless strengthen the manuscript to situate their approach within this broader body of research. Including this would allow the authors to more clearly demonstrate the advantages and added value of using the social discounting task.

3) On page 4 (lines 81-84) the authors write: "Advantages of this task over other paradigms involving resource allocation include the use of multiple traits, decisions for real recipients who vary in social closeness (rather than a single anonymous stranger), and a non-transparent task structure." It would be helpful if the authors could clarify what is meant by a "non-transparent task structure." For instance, how is the social discounting task less transparent than the SVO Slider? One might argue that participants in both tasks can readily infer that the paradigm is intended to measure prosocial/selfish behavior. Can the authors support this claim with a reference?

4) Minor: I find that the sentence spanning lines 90 to 95 is somewhat difficult to follow due to its length. The authors may consider splitting it into two shorter sentences to improve readability and clarity.

5) Minor: I believe that the statement on line 68 (i.e., "Antisociality in psychopathy and other disorders may also in part reflect deficits in executive function ...") could be supported by a more direct reference. Specifically, Burghart et al. (2024) conducted a meta-analysis showing that executive function deficits are not specific to psychopathy per se, but are more broadly associated with antisocial traits. Please note: this is a self-citation, so no need to cite it!

Burghart, M., Schmidt, S., & Mier, D. (2024). Executive functions in psychopathy: a meta-analysis of inhibition, planning, shifting, and working memory performance. *Psychological Medicine*, 54(11), 2823–2837. doi:10.1017/S0033291724001259

Methods

6) The power analysis reported on page 6 would benefit from additional details. For instance, to identify group differences on what? The cited reference by Amormino et al. is listed as "in prep," and therefore does not provide readers with accessible information for further clarification. I recommend that the authors elaborate a bit.

7) On the next page, the authors cite Berluti et al. (2024) and list it in the references as a study "in prep." However, to my knowledge, this study has already been published. If that is the case, the reference should be updated accordingly so that readers can access the study and check how the quasi-representative sample of US adults was determined.

8) It would be helpful to indicate in Table 1 which characteristics show significant differences between the two samples. While this information is provided in the text, marking these differences in the table (e.g., by using a simple asterisk) would make it easier for readers to identify key group differences at a glance.

9) Could the authors please elaborate on the purpose of administering the additional PCL:SV assessments? The manuscript states that this was done to "confirm" the high-psychopathy group, but it is unclear how confirmation was determined based on only a subsample. How were the 44 individuals selected for this additional assessment?

10) Minor: please include the statistical indices and results for the model comparisons reported on page 10 (lines 215–217).

Results

11) Minor: The number "44" should be written out as "Forty-four" (p. 12, line 264).

Discussion

12) In comparison to the rest of the manuscript, the discussion section felt somewhat underdeveloped. I would encourage the authors to move beyond simply summarizing their findings and more fully explore the broader implications of their results.

For instance, what do these findings suggest for psychopathy research conducted with student samples, particularly in light of the authors' earlier point that few studies have examined individuals with very high levels of psychopathy in community samples? Additionally, the abstract mentions that the findings may have implications for treatment, but this point is only briefly noted in the final sentence of the discussion.

Another area that warrants further interpretation is the role of antisocial behavior. The authors found that antisocial behavior in psychopathy is partly explained by increased social discounting. This raises an important conceptual question: Should antisocial behavior be considered a defining feature of psychopathy, or rather an outcome of it? This issue aligns closely with the Triarchic Model of Psychopathy and would be a valuable topic for discussion.

These are just a few suggestions, but I believe that by expanding their discussion, the authors could increase the overall impact of their manuscript.

Data and analysis scripts

I appreciate that the authors have made all data and code publicly available. I reviewed everything and was able to reconstruct the cleaned dataset and all findings as reported. That said, some parts of the code (particularly the data cleaning script) were somewhat difficult to follow and could potentially be streamlined. Additionally, I encountered a minor issue with font rendering in the figures when running the main analysis script, though this was easily resolved and may have been specific to my system. However, I do not expect the authors to revise their code, as everything was reproducible as reported. Thank you for your transparency and commitment to open science!

Speaking of transparency, I would like to note that I have suggested one of my own publications as an additional reference for this manuscript. To maintain openness, I am therefore signing this review with my name.
Matthias Burghart

Version 1:

Decision Letter:

Dear Ms Nero,

Your manuscript titled "Severity of Psychopathy in a Community-Recruited Sample is Indexed by Increased Social Discounting" has now been seen by our reviewers, whose comments appear below. In light of their advice I am delighted to say that we are happy, in principle, to publish a suitably revised version in Communications Psychology.

We therefore invite you to revise your paper one last time to address the remaining concerns of our reviewers and a list of editorial requests. At the same time we ask that you edit your manuscript to comply with our format requirements and to maximise the accessibility and therefore the impact of your work.

EDITORIAL REQUESTS:

SUBMISSION INFORMATION:

OPEN ACCESS:

Communications Psychology is a fully open access journal. Articles are made freely accessible on publication. For further information about article processing charges, open access funding, and advice and support from Nature Research, please visit <https://www.nature.com/commpsychol/open-access>

* **DATA AVAILABILITY:**

Link Redacted

Best regards,

Troy Lui, on behalf of

Inti Brazil

Troy Lui, PhD
Associate Editor
Communications Psychology

Inti Brazil, PhD
Editorial Board Member
Communications Psychology
orcid.org/0000-0001-5824-0902

REVIEWERS' COMMENTS:

Reviewer #1 (Remarks to the Author):

The authors have responded thoroughly to prior comments, and I appreciate the revisions made. This, I believe, will be a valuable contribution to the field.

One consideration for future publications is to place greater emphasis in the discussion section on how the findings align with, or diverge from, the contemporary literature. While additional citations were included, they currently appear somewhat appended rather than fully integrated. More explicit engagement with contemporary work would help situate the study in its nuanced context and highlight how these results inform interpretation and the design of future experiments.

At present, the discussion focuses primarily on points of agreement in the literature and the strengths of the study. A richer analysis that also acknowledges contrasting findings and explores their implications would elevate the discussion further.

One Steinbeis article I was thinking about - but not necessary to include - is here
<https://doi.org/10.1016/j.copsyc.2017.08.012>

Reviewer #2 (Remarks to the Author):

I am satisfied that the authors have addressed all the comments raised.

Reviewer #3 (Remarks to the Author):

All my comments have been thoroughly addressed by the authors. I look forward to reading the published paper!

Response to reviewers

REVIEWER REPORTS:

Reviewer #1 (Remarks to the Author):

Thank you for the opportunity to review this manuscript, which investigates the relationship between psychopathic traits and social discounting in a large, community-recruited sample. This is a timely and important contribution, especially given the need for research on non-institutionalized individuals with high psychopathy. The manuscript is generally well-written, the analyses are clearly presented, and the study is methodologically ambitious.

We sincerely appreciate this encouraging feedback!

That said, I offer several suggestions to enhance the clarity, contextual grounding, and methodological rigor of the paper. These are detailed below.

Citations and Framing the Contribution

The authors could enhance the impact and contextual clarity of their manuscript by expanding their citation practices to better reflect recent research and include a broader range of voices. A significant proportion of the cited literature—approximately 60%—is over a decade old. While many of these are foundational papers that rightly inform the theoretical foundation, the field has evolved substantially with more nuanced findings from work that has been testing ideas from these > 10 year old ideas that are highly relevant to the present work.

For instance, recent contributions by Nikolaus Steinbeis, who explores cognitive and neural mechanisms of prosocial and antisocial behavior, and Joseph Sakai, who has developed paradigms measuring prosocial behavior toward distant others in relation to psychopathic traits, offer frameworks that could enrich the current manuscript's discussion. Including these perspectives would not only align the manuscript with the current state of the field but would also provide readers with a more comprehensive understanding of how this study advances existing knowledge.

This adjustment would support a more inclusive and up-to-date discussion and offer the opportunity to situate the findings within current debates in the literature on psychopathy and prosocial cognition.

Thank you for this suggestion. We confirmed that 50% of our citations were indeed more than 10 years old, and have gone through and replaced many of them with updated citations, endeavoring to reflect the diverse laboratories and investigators studying psychopathy. New citations are highlighted in the text and the References section.

We agree that the specific authors highlighted by the reviewer have done important work related to our research and now cite two recent papers (Winters & Sakai, 2023; Winters et al., 2025) that highlight the role of cognitive control and affective theory of mind in psychopathy in our

literature review. We also cite Sakai et al. (2019) in our added text discussing prior studies investigating prosocial behaviors in psychopathy.

Our searches of relevant work by Steinbeis largely yielded (excellent) papers related to the neurodevelopment of prosociality in children; given our research does not directly touch on these topics we were unable to integrate relevant citations but would be glad to do so if the reviewer had a specific reference in mind.

Unpublished Citation

The manuscript cites an "in prep" manuscript:

Berluti, K., Ploe, M., Doherty, H., Jones, D., Patrick, C., Marsh, A.A. (2024). Prevalence and correlates of psychopathy in the general population. In prep.

Because there is no accessible preprint or publication associated with this reference, it is not possible to verify or evaluate the cited findings. Given that "in preparation" manuscripts are inherently unstable and may change substantially prior to publication, I recommend that this citation either be removed or replaced with a citable preprint if available.

Thank you for pointing this out! This paper is now published and we have now updated the citation:

Berluti, K., Ploe, M. L., Doherty, H., Jones, D. N., Patrick, C. J., & Marsh, A. A. (2025). Prevalence and Correlates of Psychopathy in the General Population. *Journal of Personality Disorders*, 39(1)(1–21). <https://doi.org/10.1521/pedi.2025.39.1.1>

Study Design and Methodology

The study employs an observational design using a convenience sample without random assignment. While this approach may be appropriate for the population of interest, it does necessitate a cautious interpretation of causal claims. I strongly encourage the authors to include a more substantial discussion in the limitations section addressing how the lack of randomization and self-selection may impact internal validity.

We agree with the reviewer that because it is impossible to randomly assign participants to a clinical condition (such as having high or low psychopathy) or to high or low levels of social discounting, it is important to refrain from making causal claims about the origins of the observed associations. We have ensured our manuscript does not make such claims about our findings and discuss this issue in our Limitations section.

We also discuss in this section that because we employed purposive sampling, as is typical for clinical research studies or studies of other special populations (Ahmed, 2024), our results may not be generalizable to all high-psychopathy adults in the community.

We note that because discounting tasks assess the effects of manipulating social (or temporal, etc.) distance on participants' behavior using a within-subjects design, they are not typically described as observational paradigms, although survey measures such as we used certainly are.

I note the authors tried to get ahead of review comments by talking about how self-selection is an issue but I do not feel what is provided in this context is sufficient – as this is the crux for reliance on the outcomes of this work and is a very probable area of bias.

We agree that the issue of self-selection into psychology research studies is an important one to take seriously, as it is pervasive in psychology research. Most of the population does not take part in psychology studies, and prior studies have found those who self-select into research tend to be disproportionately female, high-SES, prosocial, and exhibit higher levels of psychopathology (van Lange et al., 2011; Kaźmierczak et al., 2023; Stone et al., 2024). A recent meta-analysis of psychopathy as assessed by the TriPM (Sleep et al., 2019), which we now cite in our manuscripts, observed that the specific sample studied (e.g., undergraduates, incarcerated samples) moderated the nature of the observed associations.

A focus on studying psychopathy in incarcerated samples and undergraduate students has made drawing conclusions about psychopathy in adults in the community challenging. Neither group is representative of the broader population, and within both groups, only a small fraction of eligible respondents participates.

As we describe in our manuscript, our sample benefited from greater socioeconomic, age, geographic, educational, ethnic, racial, and gender diversity than most samples assessed for psychopathy. However, because this sample was primarily recruited from adults who sought information about psychopathy online, and because we do take seriously the issue of potential biases introduced by self-selection, we did compare results whether including or excluding high-psychopathy CloudResearch panel participants, and found that results in both cases were extremely similar.

In addition, we have now conducted analyses aimed at comparing our high-psychopathy sample to the sample used in a recent study (Berluti et al., 2025) which is the first to assess psychopathy in a quasi-representative sample of US adults recruited via Qualtrics panel and which, like the present study, used the Triarchic Psychopathy Measure to predict antisocial behavior using the Subtypes of Antisocial Behavior Questionnaire. We replicated the prior study's analyses of the association between psychopathy and antisocial behavior while controlling for age, gender, race, and household income. Results yielded the same associations, most of which were extremely similar in magnitude across the samples, e.g., between total psychopathy and antisocial behavior ($b = 0.55$, as compared to $b = 0.50$ in Berluti et al), between antisocial behavior and disinhibition ($b = 0.80$, as compared to $b = 0.88$), and between antisocial behavior and boldness ($b = -0.03$, as compared to $b = 0.03$), and between antisocial behavior and meanness ($b = 0.68$, as compared to $b = 0.33$).

Taken together, these largely consistent patterns between our sample and the only study of psychopathy to our knowledge that recruited a quasi-representative sample of the US adult population lend support to our findings being reasonably generalizable.

Key point: The paper acknowledges demographic differences between groups (age, education, fluid intelligence), which were controlled for in statistical models. However, the observed differences in core covariates associated both with group membership and outcomes (e.g., age

and cognitive ability) raise the potential for collider bias, which can compromise the validity of group comparisons.

To strengthen inference, I suggest the authors consider:

- Propensity score matching to reduce imbalance across covariates
- Sensitivity or robustness checks using permutation testing to assess significance under exchangeability
- Cross-validation procedures to assess the stability of group-level effects

Such steps would provide more robust support for the reported associations and increase confidence in the findings.

Ultimately, these issues will impact confidence in the results regardless and I believe is an important reason to scale back the language. As of now it sounds quite causal and undermines the substantial design issues of the work – please consider substantially revising the language of this article and the title to be more commensurate with the quality of evidence indicative of the design.

We appreciate these suggestions. We have now implemented propensity score matching and cross-validation as robustness checks to reduce imbalance across key covariates, including age, gender, household income, and fluid intelligence. Specifically, we used nearest-neighbor matching based on logistic regression propensity scores, achieving substantially improved covariate balance across the matched sample. Additionally, we conducted a 10-fold cross-validation procedure on the original dataset while controlling for age, gender, household income, and fluid intelligence.

We then re-ran our primary analyses following these robustness checks and report our results using propensity score matching and cross-validation in the Supplementary Materials. Effect sizes remained extremely consistent with those reported in the main manuscript, increasing our confidence that our results did not reflect collider bias or imbalances in group compositions. Notably, the consistency of results held even when using the propensity-matched sample, which involved a much reduced sample size ($N = 476$ (238 per group) versus $N = 727$ in our planned analyses) and therefore dramatically decreased statistical power.

	No robustness checks	Cross-validation	Propensity-score matching
Psychopathy / logk association (group)	$B = 0.38, p < .001,$ 95% CI [0.31, 0.45];	$B = 0.38, p < .001$ RMSE = 1.83, SD = 0.08	$B = 0.36, p < .001,$ 95% CI [0.27, 0.44]
Psychopathy / logk association (continuous)	$B = 0.40, p < .001,$ 95% CI [0.33, 0.47]	$B = 0.40, p < .001,$ RMSE = 1.82, SD = 0.07	$B = 0.36, p < .001,$ 95% CI [0.27, 0.44]
Meanness / logk	$B = .41, p < .001,$ 95% CI [0.29, 0.54]	$B = 0.41, p < .001,$ RMSE = 1.80, SD = 0.08	$B = .41, p < .001,$ 95% CI [0.26, 0.56]

Covariate Selection and Control Strategy

The manuscript includes demographic covariates (age, gender, household income, and fluid intelligence) in all models, which is a good practice. However, a more theory-driven approach to selecting covariates could be valuable. Have the authors considered including additional covariates based on theoretical relevance (e.g., childhood adversity, SES beyond income, other confounding mental health symptoms)? Alternatively, including a statement clarifying the rationale for the chosen set would be helpful. Acknowledging the limitations of the current control strategy would also strengthen the discussion.

We regret that our manuscript was not clearer about the theory-driven selection of our covariates. We now explicitly state in our Introduction that we controlled for age, household income, and fluid intelligence because these three variables are consistently associated with population-level differences in generosity (*age*: Lockwood et al., 2021; *wealth/income*: Vanags et al., 2025, Kushlev et al., 2022, Wu et al., 2025; *intelligence*: Böckler et al., 2016; Guo et al., 2019). We also viewed income as important to include because this is a monetary task. We also included age and cognitive ability because these variables differed across groups. In terms of assessing SES, we focused on household income because this is one of the most commonly used quantitative indices of SES (Conway et al., 2019), and is reliably linked to various indices of generosity.

We now report in our Introduction (p. 6):

“We also considered potential effects of age, gender, socio-economic status (household income), and cognitive intelligence, given prior evidence that increases in these variables are reliably associated with increased generosity (Liebe et al., 2022; Lockwood et al., 2021; Böckler et al., 2016).”

Retrospectively reported child adversity can be a difficult variable to interpret, as it may be closely associated with current psychopathology than with concurrently measured childhood adversity (Danese & Wisdom, 2020). We therefore did not include this variable in our regressions.

We did not collect data on total wealth, parental education, childhood household income, or housing status and note this in our Limitations (p. 20):

“We evaluated the potential role of covariates such as age, gender, household income, and fluid intelligence in social discounting, but other potentially relevant factors not measured here could include housing status, early-life adversity, or childhood household income.”

Sample Validity and Data Quality

The study combines data from two distinct sources: one group recruited through CloudResearch and another via a non-profit website offering psychopathy resources. While the use of community samples is a strength, these recruitment methods differ significantly. CloudResearch applies quality controls for online respondents, while the website group is self-selected and recruited via a psychopathy-specific context.

It would benefit the manuscript to include additional discussion (or ideally analyses) addressing:

- The potential for sampling bias in the self-recruited group

We appreciate the reviewer’s emphasis on the potential for sampling bias in our self-recruited group. In response to this concern, we now state in our Limitations section (p. 19):

“First, like many clinical research studies and other studies recruiting special populations (Andrade, 2021), this study employed a cross-sectional, correlational design using a purposive sample. We thus refrain from drawing causal conclusions about the origin of the observed effects. Because most of our high-psychopathy participants were recruited after seeking information about psychopathy online, they were in part self-selected. Self-selection is a pervasive consideration in psychological research, as research participation is voluntary even in institutional and clinical settings, such that research participants represent a small fraction of potentially eligible adults, and may generally be biased toward populations that are disproportionately female, wealthy, and/or prosocial (van Lange et al., 2011; Kaźmierczak et al., 2023; Stone et al., 2024). A meta-analysis of psychopathy as assessed by the TriPM (Sleep et al., 2019) observed that the specific sample studied (e.g., undergraduates, imprisoned samples) moderated the nature of the observed associations. Lending some support to the generalizability of our results, we observed similar results whether including or excluding high-psychopathy CloudResearch panel participants. We also observed high and consistent self-reported commitment and response reliability across participant groups. However, our sample may nonetheless be non-representative of high-psychopathy adults in the community.”

- Whether participant characteristics (e.g., response patterns, reliability) differ by recruitment method
- Any efforts to ensure response validity across groups (e.g., attention checks, repeated measures, or validation items)

The paper notes exclusion of participants failing multiple attention checks, which is reassuring. However, it would be helpful to report any data quality comparisons across recruitment pools and acknowledge the limitations of online, unsupervised testing environments, particularly for a psychopathy sample.

The authors should also consider identifying any patterned responses in the data that may indicate a less reliable way or reporting that is not in good faith and control for this. These are online samples and anything the authors can do to make readers more confident the data is handled appropriately would improve the impact of this work.

We appreciate this concern and now report the multiple steps we have taken to assess the reliability and validity of responding across groups.

Following recently generated recommendations for maximizing the quality of online data, in the beginning of the survey we included a commitment request question (Geisen, 2022; Hibben et al., 2022): “Do you commit to providing thoughtful answers to the questions in this survey”. Participants could respond with “Yes”, “I can’t promise”, or “No”, with participants excluded for

selecting “No.” No participants in either participant group selected “No”. We compared the proportion of participants who responded “I can’t promise” versus “Yes” across recruitment sources (1.7% from psychopathy online platform; 0.28% from CloudResearch) and found no group differences, $\chi^2(1) = 2.34, p = .13$.

As the reviewer notes, we then excluded participants in either group if they failed 2 or more of the 4 attention checks. This resulted in the exclusion of 10 participants in the high psychopathy group and 2 in the Cloud Research group. Thus, all included participants passed at least 3 of 4 attention checks.

We also assessed survey response times for the remaining participants in both groups and found the high-psychopathy group exhibited higher response times than controls, $t(331.63) = -2.67, p = .008$, which is the opposite of what would be expected were the high-psychopathy group not responding in good faith.

Finally, across both groups, we assessed the reliability of responses on standard measures and found very high reliability for both the high-psychopathy group (TriPM total: $\alpha = .83$, 95% CI [.79 0.85]; TriPM Meanness: $\alpha = 0.82$ [0.79, 0.85]; TriPM Disinhibition: $\alpha = 0.80$, 95% CI [0.77, 0.83]; TriPM Boldness: $\alpha = 0.82$, 95% CI [0.79, 0.85], STAB: $\alpha = .91$, 95% CI [0.89, 0.93]) and CloudResearch-recruited group (TriPM total: $\alpha = .89$, 95% CI [.88 0.90]; TriPM Meanness: $\alpha = 0.89$, 95% CI [0.888, 0.91]; TriPM Disinhibition: $\alpha = 0.89$, 95% CI [0.88, 0.91]; TriPM Boldness: $\alpha = 0.88$, 95% CI [0.86, 0.89]; STAB: $\alpha = .95$, 95% CI [0.94, 0.96]). These reliability results are also inconsistent with either sample responding unreliably or not in good faith. These estimates are now reported in the Supplemental Materials.

We believe that together these metrics provide strong evidence that participants in both groups provided data of high and comparable quality. We now make note of this in our Discussion.

Transparency and Code Sharing

The authors commendably share both data and code via the OSF. However, the shared R Markdown script could be improved in clarity. Currently, the code includes dense inline commenting within code blocks, making it difficult to distinguish between executable code and explanatory text.

Best practice suggests:

- Housing detailed explanations and context ****outside**** code chunks
- Using inline comments sparingly within code (e.g., to annotate logic)
- Providing a narrative flow in markdown cells for clarity
- Provide not just the code but also the markdown HTML once it is knitted so others can see the exact output you did when conducting the analysis (as well as data and modeling assumption checks that are not reported on **see below comment**).

This would improve reproducibility and accessibility for external readers or future collaborators.

Thank you for this helpful feedback. In response, we have reorganized our data cleaning and analysis scripts to improve clarity and accessibility. Additionally, we have created a knitted pdf version of the R markdown to ensure others can view the full output of the analysis, including

model assumptions and checks. Both the updated R markdown files and the rendered PDF outputs are now available on the OSF website for this project.

Methods and Diagnostics

The manuscript clearly states that the study was not preregistered. Given the flexibility in analytic choices, a brief post hoc discussion of analytic decision points, degrees of freedom, and how multiple comparisons were handled would be helpful.

Thank you for this important point. Because clinical studies of psychopathy and social discounting tasks have been analyzed using various approaches, to address concerns about analytic flexibility and researcher degrees of freedom, we adopted a multiverse approach and report the results of all tests. We tested models using both group-based and continuous predictors (TriPM) and ran our analyses both after dropping participants outside of their recruited groups' TriPM score and transferring them to the appropriate group. Although our primary analyses of social discounting used logk, we present secondary analyses of results using AUC for completeness and to aid in future meta-analyses and replication attempts. The convergence of findings across all of these analytic paths increases confidence in the robustness and reliability of the results.

We now also incorporate multiple comparison corrections in the manuscript where appropriate. For example, when examining group differences in criminal history (committing, charged, or convicted of a crime) we applied a Bonferroni correction to account for the three related tests. When conducting similar analyses using both group-based and continuous models we also now apply Bonferroni correction. These corrections do not alter our results or conclusions.

We have updated the manuscript to better describe these analytic decisions.

It would also strengthen the manuscript if the authors elaborated on why their chosen analytic approach was preferred over more flexible modeling frameworks that could simultaneously account for both group-level and continuous variation. For instance, if I understand the analytic plan correctly, it appears the authors conducted separate analyses to address group comparisons and continuous effects, when in fact a single hierarchical or mixed-effects model could have captured both, while also improving parameter reliability by leveraging the full structure of the data.

We appreciate this point and agree that hierarchical or mixed effects models often offer a valuable framework for simultaneously estimating group-level and continuous effects. In our case, group membership was defined based on TriPM scores. Because a variable cannot be nested within itself, continuously varying TriPM scores cannot be described as nested within groups. Group and TriPM scores are also highly collinear by definition (point-biserial correlation: $r = .87$, 95% CI [0.86, 0.87], $p < .001$). Thus, including both group and TriPM scores in a single model introduces mutual adjustment: the group coefficient reflects the group differences controlling for TriPM (i.e., at the average TriPM score), and the TriPM coefficient reflects association with logk controlling for group. Interpretation becomes difficult as there is little variable in group differences that is independent of TriPM. This also introduces

multicollinearity, it becomes unclear what it means to be in the high psychopathy group once we have already accounted for the individual's TriPM score.

That said, because we appreciate reviewer's suggestion of the value of modeling both group and continuous effects of psychopathy within a unified analytic framework, we conducted a piecewise linear spline regression approach, which flexibly models psychopathy as a continuous predictor while accounting for the clinically relevant thresholds that separate the groups. Because clinical cutoff score differ across genders, we used gender-specific knots based on established TriPM cutoffs for clinically high psychopathy (Berluti et al., 2025; Male = 105; Female/Other = 91) and constructed separate B-spline basis functions for males and females. After extracting individual logk values from the hyperbolic model presented in the manuscript, the spline model allowed us to simultaneously estimate the relationship between psychopathy and logk across the full range of scores while testing whether the relationship differed below or above the cutoff for each gender. The resulting model revealed that psychopathy was associated with increased logk for females below the clinical cutoff ($B = 0.23$, 95% CI [0.07, 0.38], $p = .005$) but not for males below the clinical cutoff ($B = 0.10$, 95% CI [-0.06, 0.26], $p = .22$). In contrast, psychopathy was robustly associated with increased logk above the cutoff in both females ($B = 0.39$, 95% CI [0.32, 0.46], $p < .001$) and males ($B = 0.24$, 95% CI [0.17, 0.32], $p < .001$). This approach captures both categorical and continuous effects of psychopathy while minimizing concerns related to multicollinearity and interpretability. The spline analyses, which we include in the Supplemental Materials reinforce the findings from our primary models, providing additional confidence in the robustness of our conclusions.

Additionally, there is no mention of model diagnostics or checks for statistical assumptions (e.g., residual normality, linearity, multicollinearity). Including a brief note that assumptions were tested that is demonstrated in the code—or acknowledging this as a limitation and providing justification for continuing anyway—would be advisable.

Thank you for this suggestion. We can confirm that we checked all models for standard statistical assumptions, including residual normality, linearity, and multicollinearity. Across models, assumptions were generally met: residual distributions were approximately normal, relationships linear, and variance inflation factors (VIFs) indicated low multicollinearity. However, models assessing the relationship between age and antisocial behavior had decreased residual normality, which we now note in the manuscript. We now include the R code used to check these assumptions.

The skewness and kurtosis of our variables are reported in the supplemental materials. Most variables were within acceptable ranges; the exception being Fluid Intelligence, which exhibited slight kurtosis. However, log transformation did not improve model performance, so untransformed values were retained in the final analyses. This is now noted in Supplementary Table S5.

Importantly, we recruited participants into two distinct groups based on TriPM scores, which naturally introduces non-normality in the distribution of some variables. This recruitment strategy is another reason we report both group-based and continuous analyses, allowing us to

account for distributional deviations while examining effects through complementary analytic lenses.

Lastly, for the hyperbolic model, we have now revised the manuscript to include the Δ AIC comparisons and Akaike weights for the models considered (hyperbolic, linear, and exponential). Specifically, we report that the hyperbolic model yielded the lowest AIC and that both the exponential and linear models had Δ AIC values well above 10, indicating substantially poorer fit (Burnham & Anderson, 2002). We also added the Akaike weight of the hyperbolic model (AICcWT = 1.00) based on the results from the AICcmodavg package (Mazerolle, 2023), further supporting its selection as the best-fitting model. These details are now provided in the revised text on page 10 of the revised manuscript.

Reviewer #2 (Remarks to the Author):

Thank you for the opportunity to review this manuscript, which I very much enjoyed reading. Indeed, it is very well written and organised. It presents novel findings on a very large sample of individuals spanning low to very high psychopathic scores performing a social discounting task. I do not have any concerns about the methodology, but offer some suggestions, which could help to improve this already excellent manuscript.

Introduction

I was surprised that there were no hypotheses formulated in the last paragraph of the introduction (although the method section states that the hypotheses were not pre-registered, which is a bit confusing), as there's been previous work that is cited by the authors and clearly one would expect a positive association between psychopathic traits and social discounting.

Admittedly, it might be a bit late now to include hypotheses, but I leave the authors and the editor decide on that aspect.

Thank you for highlighting this! We did indeed predict increased social discounting in the high-psychopathy group based on our prior work and have now stated this more explicitly in the introduction (p. 6):

“Following prior work linking reduced social discounting to highly prosocial phenotypes (Rhoads et al., 2023), we predicted psychopathy would be associated with increased social discounting. We also predicted that the association between psychopathy and antisociality would be partly mediated by reductions in the subjective valuation of others' welfare as psychopathy increases.”

Relatedly, there are analyses examining age effects and moderation by age, but there is no mention in the discussion of age being a potentially important factor that would be formally tested in the manuscript; I think this should be touched on in the introduction.

We agree and now explicitly describe prior research linking age to social discounting and related variables in our introduction (p. 6).

“We also considered potential effects of age, gender, socio-economic status (household income), and cognitive intelligence, given prior evidence that increases in these variables are reliably associated with increased generosity (Liebe et al., 2022; Lockwood et al., 2021; Böckler et al., 2016).”

Methods

For the description of the power analysis, the statistical threshold should be stated (i.e., at $p < .05$ I assume?)

Thank you for alerting us to this omission; we now clarify that the statistical threshold was at $p < .05$.

For the PCL-SV description, it would be helpful to state what cut-off is considered indicative of psychopathy, which I believe is 18.

Thank you for indicating this! We have specified this cut-off score in the Methods section (p. 8):

“Although psychopathy is now agreed to be continuously distributed, scores ≥ 18 have been previously used as clinical cutoffs (Cooke et al., 1999), although scores ≥ 8 optimize specificity and sensitivity for predicting outcomes such as community violence in civil psychiatric samples (Skeem & Mulvey, 2001).”

Results

The supplementary material includes up to 44 Supplementary tables(!), but not all of those are mentioned in the main manuscript. Are all those tables really necessary, as they are not mentioned in the main ms?

Thank you for raising this point. We have now added references to the Supplementary Tables and described the accompanying analysis in the manuscript. These supplementary analyses largely report secondary or robustness checks that were conducted to ensure transparency and strengthen confidence in our findings (see responses to Reviewer 1). For example, in the main manuscript, we focus on logk as the primary dependent variable and metric of social discounting. However, following our “multiverse” approach to analyzing our data we provide supplementary analyses utilizing AUC values in our Supplement to facilitate future replication and meta-analysis efforts, (Supplementary Tables S18-S26). Similarly, we note in the main manuscript that participants falling outside the TriPM cutoff scores for their recruited group were reassigned to the appropriate group. However, to ensure the robustness of our findings (a topic of concern to Reviewer 1), we conducted parallel analysis (with logk and AUC) in which those participants were excluded entirely (Supplementary Tables S27-S48). Lastly, we added new analyses in response to comments from Reviewer 1 which are now reported in Supplementary Tables S1-S4. These additional analyses yielded consistent results but are reported separately in the supplement.

We are happy to reduce or consolidate these tables if the editorial team agrees it would improve clarity.

Also, it would be easier to read the results if all the stats were included in the table in an additional column or 2; that would make the main text of the results lighter.

Thank you for this suggestion. For the models in which tables are provided in the main text and Supplemental Materials, the model statistics, including F-statistics, were moved from the text to the tables.

p.13 This subtitle section: “Do Differences in Social Discounting Correspond to Differences in Antisocial Behavior?” the word “Corresponds” should be replaced by “Mediate” or “Account”.

We agree and have replaced “corresponds” with “mediate” to make the goal of the analyses clearer.

p.14 Mediation analysis: Please state what percentage of psychopathy-ASB association was accounted by the mediator.

Thank you for addressing this point of clarification. We have added beta values for the proportion mediated along with the confidence interval and p-value for both mediation models we present.

Figure 1: Specify what the shaded area represents in the graph.

Thank you! We have added the sentence “The shaded region represents the standard error around the mean” under the note for Figure 1.

Reviewer #3 (Remarks to the Author):

Thank you for the opportunity to review the manuscript titled “Severity of Psychopathy in a Community-Recruited Sample is Indexed by Increased Social Discounting”. Overall, the topic is highly relevant, and the manuscript is generally well-written and clearly structured. I believe it has the potential to be published in Nature Communications Psychology. That said, I have a few comments and suggestions that may help further improve the manuscript.

Abstract

1) The abstract states that the authors used a “sample of very-high psychopathy adults” (p. 2, line 32). It would be helpful to define what constitutes “very-high” psychopathy, as was done later in the manuscript on page 11 (i.e., “top 95th percentile of TriPM scores”).

We agree and now have added this parenthetical to the manuscript: (above the 95th percentile of TriPM scorers)

Introduction

2) My main concern with the introduction is the absence of a review of findings from studies examining psychopathy using economic games and/or the Social Value Orientation (SVO) Slider. Although these paradigms differ from the social discounting task (as the authors note), it would nonetheless strengthen the manuscript to situate their approach within this broader body of research. Including this would allow the authors to more clearly demonstrate the advantages and added value of using the social discounting task.

We agree that this body of literature is relevant and have added text about the SVO and related tasks (e.g., the Dictator Game) and how the Social Discounting task differs from these tasks (p. 5):

“Because the majority of actual prosocial behavior is aimed at benefiting close others rather than strangers (Cialdini et al., 1997; Curry et al., 2013; Stewart-Williams, 2007), this task also benefits from increased ecological validity relative to other commonly-used paradigms such as the dictator game and social value orientation task (SVO) that focus on generosity toward a single anonymous stranger and which were created to assess other constructs (for example, individualistic versus competitive, altruistic, or cooperative outcomes in the case of the SVO). As a result of these task features, the social discounting task has higher predictive validity than other prosocial tasks, including the SVO and dictator game (Böckler et al., 2016) or self-report measures (Vekaria et al., 2017; Rhoads et al., 2023). Furthermore, neuroimaging and behavioral research support the conclusion that choices during the task reflect variation in the subjective valuation of others’ welfare rather than effortful suppression of selfish responses (Rhoads et al., 2023).”

3) On page 4 (lines 81-84) the authors write: “Advantages of this task over other paradigms involving resource allocation include the use of multiple traits, decisions for real recipients who vary in social closeness (rather than a single anonymous stranger), and a non-transparent task structure.” It would be helpful if the authors could clarify what is meant by a “non-transparent task structure.” For instance, how is the social discounting task less transparent than the SVO Slider? One might argue that participants in both tasks can readily infer that the paradigm is intended to measure prosocial/selfish behavior. Can the authors support this claim with a reference?

We appreciate the reviewer’s feedback on this point. The non-transparent task structure of the social discounting task is, we agree, primarily in contrast to the dictator game. We have revised this section as described above to emphasize that the Social Discounting Task embeds prosocial decisions within a structured social context, which, in addition to the inclusion of multiple trials per recipient, may improve both ecological and predictive validity.

4) Minor: I find that the sentence spanning lines 90 to 95 is somewhat difficult to follow due to its length. The authors may consider splitting it into two shorter sentences to improve readability and clarity.

Thank you for this suggestion; we have now separated this sentence into two sentences.

5) Minor: I believe that the statement on line 68 (i.e., “Antisociality in psychopathy and other disorders may also in part reflect deficits in executive function ...”) could be supported by a more direct reference. Specifically, Burghart et al. (2024) conducted a meta-analysis showing that executive function deficits are not specific to psychopathy per se, but are more broadly associated with antisocial traits. Please note: this is a self-citation, so no need to cite it!

Burghart, M., Schmidt, S., & Mier, D. (2024). Executive functions in psychopathy: a meta-analysis of inhibition, planning, shifting, and working memory performance. *Psychological Medicine*, 54(11), 2823–2837. doi:10.1017/S0033291724001259

Thank you for this suggestion. We have added Burghart et al., 2024 to add additional support to that claim, as we believe it is a reference that readers will value.

Methods

6) The power analysis reported on page 6 would benefit from additional details. For instance, to identify group differences on what? The cited reference by Amormino et al. is listed as “in prep,” and therefore does not provide readers with accessible information for further clarification. I recommend that the authors elaborate a bit.

Thank you for this suggestion. We have clarified in the manuscript that the power analysis was conducted to identify group differences in social discounting.

The previously cited manuscript is now in revision and a preprint is publicly available: https://papers.ssrn.com/sol3/papers.cfm?abstract_id=5179064. We have updated the citation accordingly (Amormino et al., *In Revision*).

7) On the next page, the authors cite Berluti et al. (2024) and list it in the references as a study “in prep.” However, to my knowledge, this study has already been published. If that is the case, the reference should be updated accordingly so that readers can access the study and check how the quasi-representative sample of US adults was determined.

The reviewer is correct—this paper is now published, and we have now provided the following citation in the manuscript accordingly:

Berluti, K., Ploe, M. L., Doherty, H., Jones, D. N., Patrick, C. J., & Marsh, A. A. (2025). Prevalence and Correlates of Psychopathy in the General Population. *Journal of Personality Disorders*, 39(1)(1–21). <https://doi.org/10.1521/pedi.2025.39.1.1>

8) It would be helpful to indicate in Table 1 which characteristics show significant differences between the two samples. While this information is provided in the text, marking these differences in the table (e.g., by using a simple asterisk) would make it easier for readers to identify key group differences at a glance.

Thank you for this suggestion. A new column was added to Table 1 indicating p-values associated with group differences for each variable.

9) Could the authors please elaborate on the purpose of administering the additional PCL:SV assessments? The manuscript states that this was done to “confirm” the high-psychopathy group, but it is unclear how confirmation was determined based on only a subsample. How were the 44 individuals selected for this additional assessment?

Because this study was conducted online, there is always some concern about veridical responding and participant characteristics (Douglas et al., 2023, Newman et al., 2021), as discussed by Reviewer 1. In addition, different measures of psychopathy capture different features of the construct to different degrees (Evans & Tully, 2016; Patrick, 2022). Thus, it is common to use multiple measures of psychopathy to assess the construct (e.g., Bowes et al., 2019, Costello et al., 2019; Marsh et al., 2008). To support our claims of elevated levels of psychopathy in our high-psychopathy sample, we conducted interviews with a subset of participants ($n = 44$) and assessed psychopathy using the PCL:SV. These respondents were invited on a rolling basis as they completed the initial survey and indicated willingness to be recontacted until we reached our goal of 15% of high psychopathy participants. The robust positive correlations between PCL:SV scores and TriPM scores and subscale scores, and high PCL:SV scores in high-psychopathy group (Coid et al., 2009) increases our confidence in our high-psychopathy sample.

10) Minor: please include the statistical indices and results for the model comparisons reported on page 10 (lines 215–217).

Thank you for this suggestion. We now include Δ AIC comparisons and Akaike weights for the models (hyperbolic, linear, and exponential). The hyperbolic model yielded the lowest AIC whereas the exponential and linear models had Δ AIC values well above 10, indicating poorer fit (Burnham & Anderson, 2002). We also calculated the Akaike weight of the hyperbolic model (AICcWT = 1.00) using the AICcmodavg package (Mazerolle, 2023), further supporting its selection as the best-fitting model. These details are now included on page 10.

Results

11) Minor: The number “44” should be written out as “Forty-four” (p. 12, line 264).

Thank you for noting this. We have changed “44” to “Forty-four”.

Discussion

12) In comparison to the rest of the manuscript, the discussion section felt somewhat underdeveloped. I would encourage the authors to move beyond simply summarizing their findings and more fully explore the broader implications of their results.

For instance, what do these findings suggest for psychopathy research conducted with student samples, particularly in light of the authors’ earlier point that few studies have examined individuals with very high levels of psychopathy in community samples? Additionally, the abstract mentions that the findings may have implications for treatment, but this point is only briefly noted in the final sentence of the discussion.

We thank the reviewer for this suggestion. We have now revised the discussion to better elaborate on the broader implications of our findings as follows:

“These findings have implications for psychopathy research conducted in student samples, as undergraduate participants may score lower on psychopathy dimensions and exhibit restricted variance. As a result, effects observed in community or clinical samples—such as the strong relationship between social discounting and antisocial behavior—may be attenuated in student samples (Sleep et al., 2019). More explicit efforts by researchers to identify similarities and differences between observed patterns in clinical and subclinical psychopathy may be valuable. These findings also have potential implications for intervention and treatment. If reduced valuation of others’ welfare partly accounts for increased antisocial behavior in psychopathy, as our mediation analysis suggests, treatments targeting social valuation processes may help reduce harm in people with psychopathy. For example, interventions designed to enhance the representations of others’ needs relative to one’s own could be effective in decreasing antisocial behavior.” (p. 19)

Another area that warrants further interpretation is the role of antisocial behavior. The authors found that antisocial behavior in psychopathy is partly explained by increased social discounting. This raises an important conceptual question: Should antisocial behavior be considered a defining feature of psychopathy, or rather an outcome of it? This issue aligns closely with the Triarchic Model of Psychopathy and would be a valuable topic for discussion. These are just a few suggestions, but I believe that by expanding their discussion, the authors could increase the overall impact of their manuscript.

Thank you for raising this point. We have added a new paragraph to the discussion to address this question:

“Our results also potentially speak to the question of whether antisocial behaviors, including criminal behaviors, are an intrinsic feature versus a downstream correlate of psychopathy. Psychopathy is thought to be a personality construct that reflects several sub-components that vary continuously across the population (Berluti et al., 2025; Sellbom & Drislane, 2021) and that include traits like meanness and narcissism that indicate devaluation of others’ welfare. These traits are robust predictors of antisocial behavior (Berluti et al., 2025; Bergström & Farrington, 2022; Virtanen et al., 2022). Our results suggest that antisocial behaviors that reduce others’ welfare may be intrinsically potentiated by very low subjective valuation of others’ welfare in high-psychopathy adults, even when psychopathy is assessed using triarchic measures like the TriPM that de-emphasize criminal and antisocial behavior relative to PCL-based assessments (Patrick, 2009; Hare, 2003). This may reflect the close association between devaluation of others’ welfare and the meanness subscale, which is a core feature of all major psychopathy measures.”

Data and analysis scripts

I appreciate that the authors have made all data and code publicly available. I reviewed everything and was able to reconstruct the cleaned dataset and all findings as reported. That said, some parts of the code (particularly the data cleaning script) were somewhat difficult to follow and could potentially be streamlined. Additionally, I encountered a minor issue with font rendering in the figures when running the main analysis script, though this was easily resolved

and may have been specific to my system. However, I do not expect the authors to revise their code, as everything was reproducible as reported. Thank you for your transparency and commitment to open science!

We appreciate the reviewer taking the time to review our data cleaning and analysis scripts! Thank you for confirming the data reproducibility of the current study. We have fixed the font rendering in the figures produced in the data analysis script. Additionally, we have reorganized our data cleaning and analysis scripts to improve clarity and accessibility. Lastly, we have created a knitted pdf version of the R markdown to ensure others can view the full output of the analysis, including model assumptions and checks. Both the updated R markdown files and the rendered PDF outputs are now available on the OSF website for this project.

Speaking of transparency, I would like to note that I have suggested one of my own publications as an additional reference for this manuscript. To maintain openness, I am therefore signing this review with my name.

Matthias Burghart

Thank you again for your thoughtful and constructive feedback on our manuscript, Dr. Burghart.

References

- Ahmed, S. K. (2024). How to choose a sampling technique and determine sample size for research: A simplified guide for researchers. *Oral Oncology Reports, 12*, 100662. <https://doi.org/10.1016/j.oor.2024.100662>
- Bergstrøm, H., & Farrington, D. P. (2022). Psychopathic personality and criminal violence across the life-course in a prospective longitudinal study: Does psychopathic personality predict violence when controlling for other risk factors? *Journal of Criminal Justice, 80*, 101817. <https://doi.org/10.1016/j.jcrimjus.2021.101817>
- Berluti, K., Ploe, M. L., Doherty, H., Jones, D. N., Patrick, C. J., & Marsh, A. A. (2025). Prevalence and Correlates of Psychopathy in the General Population. *Journal of Personality Disorders, 39(1)*(1–21). <https://doi.org/10.1521/pedi.2025.39.1.1>
- Böckler, A., Tusche, A., & Singer, T. (2016). The structure of human prosociality: Differentiating altruistically motivated, norm motivated, strategically motivated, and self-reported prosocial behavior. *Social Psychological and Personality Science, 7(6)*, 530–541. <https://doi.org/10.1177/1948550616639650>
- Bowes, S. M., Watts, A. L., Thompson, W. W., & Lilienfeld, S. O. (2019). Clarifying the association between psychopathy dimensions and internalizing symptoms in two community samples: The role of general personality. *Personality and Individual Differences, 147*, 144–155. <https://doi.org/10.1016/j.paid.2019.04.024>
- Burghart, M., Schmidt, S., & Mier, D. (2024). Executive functions in psychopathy: a meta-analysis of inhibition, planning, shifting, and working memory performance. *Psychological Medicine, 54(11)*, 2823–2837. doi:10.1017/S0033291724001259

- Burnham, K. P., & Anderson, D. R. (2004). Multimodel Inference: Understanding AIC and BIC in Model Selection. *Sociological Methods & Research*, 33(2), 261–304.
<https://doi.org/10.1177/0049124104268644>
- Cialdini, R. B., Brown, S. L., Lewis, B. P., Luce, C., & Neuberg, S. L. (1997). Reinterpreting the empathy–altruism relationship: When one into one equals oneness. *Journal of Personality and Social Psychology*, 73(3), 481–494. <https://doi.org/10.1037/0022-3514.73.3.481>
- Coid, J., Yang, M., Ullrich, S., Roberts, A., & Hare, R. D. (2009). Prevalence and correlates of psychopathic traits in the household population of Great Britain. *International journal of law and psychiatry*, 32(2), 65–73. <https://doi.org/10.1016/j.ijlp.2009.01.002>
- Conway, D. I., McMahon, A. D., Brown, D., & Leyland, A. H. (2019). Measuring socioeconomic status and inequalities. In S. Vaccarella, J. Lortet-Tieulent, R. Saracci, D. I. Conway, K. Straif, & C. P. Wild (Eds.), *Reducing social inequalities in cancer: Evidence and priorities for research*. International Agency for Research on Cancer.
<http://www.ncbi.nlm.nih.gov/books/NBK566205/>
- Cooke, D., Michie, C., Hart, S. D., & Hare, R. D. (1999). Evaluating the Screening Version of the Hare Psychopathy Checklist—Revised (PCL:SV): An item response theory analysis. *Psychological Assessment*, 11(1), 3–13.
- Costello, T. H., Smith, S. F., Bowes, S. M., Riley, S., Berns, G. S., & Lilienfeld, S. O. (2019). Risky business: Psychopathy, framing effects, and financial outcomes. *Journal of Research in Personality*, 78, 125–132. <https://doi.org/10.1016/j.jrp.2018.11.006>

- Curry, O., Roberts, S. G. B., & Dunbar, R. I. M. (2013). Altruism in social networks: Evidence for a 'kinship premium.' *British Journal of Psychology*, *104*(2), 283–295.
<https://doi.org/10.1111/j.2044-8295.2012.02119.x>
- Danese, A., & Widom, C. S. (2020). Objective and subjective experiences of child maltreatment and their relationships with psychopathology. *Nature Human Behaviour*, *4*(8), 811–818.
<https://doi.org/10.1038/s41562-020-0880-3>
- Douglas, B. D., Ewell, P. J., & Brauer, M. (2023). Data quality in online human-subjects research: Comparisons between MTurk, Prolific, CloudResearch, Qualtrics, and SONA. *PloS one*, *18*(3), e0279720. <https://doi.org/10.1371/journal.pone.0279720>
- Evans, L., & Tully, R. J. (2016). The triarchic psychopathy measure (TriPM): Alternative to the PCL-R?. *Aggression and Violent Behavior*, *27*, 79-86.
- Geisen, E. (2022, August 4). *Using Attention Checks in Your Surveys May Harm Data Quality*. Qualtrics. <https://www.qualtrics.com/blog/attention-checks-and-data-quality/>
- Guo, Q., Sun, P., Cai, M., Zhang, X., & Song, K. (2019). Why are smarter individuals more prosocial? A study on the mediating roles of empathy and moral identity. *Intelligence*, *75*, 1–8. <https://doi.org/10.1016/j.intell.2019.02.006>
- Hare, R. D. (2003). *The Hare Psychopathy Checklist— Revised* (2nd ed.). Toronto: Multi-Health Systems.
- Hibben, K. C., Felderer, B., & Conrad, F. G. (2022). Respondent commitment: Applying techniques from face-to-face interviewing to online collection of employment data. *International Journal of Social Research Methodology*, *25*(1), 15–27.
<https://doi.org/10.1080/13645579.2020.1826647>

- Kaźmierczak, I., Zajenkowska, A., Rogoza, R., Jonason, P. K., & Ścigała, D. (2023). Self-selection biases in psychological studies: Personality and affective disorders are prevalent among participants. *PLOS ONE*, *18*(3), e0281046. <https://doi.org/10.1371/journal.pone.0281046>
- Kushlev, K., Radosic, N., & Diener, E. (2022). Subjective well-being and prosociality around the globe: Happy people give more of their time and money to others. *Social Psychological and Personality Science*, *13*(4), 849–861. <https://doi.org/10.1177/19485506211043379>
- Liebe, U., Schwitter, N., & Tutić, A. (2022). Individuals of high socioeconomic status are altruistic in sharing money but egoistic in sharing time. *Scientific Reports*, *12*(1), 10831. <https://doi.org/10.1038/s41598-022-14800-y>
- Lockwood, P. L., Abdurahman, A., Gabay, A. S., Drew, D., Tamm, M., Husain, M., & Apps, M. A. J. (2021). Aging Increases Prosocial Motivation for Effort. *Psychological Science*, *32*(5), 668–681. <https://doi.org/10.1177/0956797620975781>
- Marsh, A. A., Finger, E. C., Mitchell, D. G. V., Reid, M. E., Sims, C., Kosson, D. S., Towbin, K. E., Leibenluft, E., Pine, D. S., & Blair, R. J. R. (2008). Reduced amygdala response to fearful expressions in children and adolescents with callous-unemotional traits and disruptive behavior disorders. *American Journal of Psychiatry*, *165*(6), 712–720. <https://doi.org/10.1176/appi.ajp.2007.07071145>
- Mazerolle, M.J. (2023). `_AICcmodavg`: Model selection and multimodel inference based on (Q)AIC(c). R package version 2.3.3, <<https://cran.r-project.org/package=AICcmodavg>>.

- Newman, A., Bavik, Y. L., Mount, M., & Shao, B. (2021). Data collection via online platforms: Challenges and recommendations for future research. *Applied Psychology, 70*(3), 1380-1402.
- Patrick, C. J. (2022). Psychopathy: Current knowledge and future directions. *Annual Review of Clinical Psychology, 18*(1), 387–415. <https://doi.org/10.1146/annurev-clinpsy-072720-012851>
- Patrick, C. J., Fowles, D. C., & Krueger, R. F. (2009). Triarchic conceptualization of psychopathy: Developmental origins of disinhibition, boldness, and meanness. *Development and Psychopathology, 21*(3), 913–938. <https://doi.org/10.1017/S0954579409000492>
- Rhoads, S. A., O’Connell, K., Berluti, K., Ploe, M. L., Elizabeth, H. S., Amormino, P., Li, J. L., Dutton, M. A., VanMeter, A. S., & Marsh, A. A. (2023). Neural responses underlying extraordinary altruists’ generosity for socially distant others. *PNAS Nexus, 2*(7), pgad199. <https://doi.org/10.1093/pnasnexus/pgad199>
- Sakai, J. T., Raymond, K. M., McWilliams, S. K., & Mikulich-Gilbertson, S. K. (2019). Testing helping behavior and its relationship to antisocial personality and psychopathic traits. *Psychiatry Research, 274*, 98–104. <https://doi.org/10.1016/j.psychres.2019.02.022>
- Sellbom, M., & Drislane, L. E. (2021). The classification of psychopathy. *Aggression and Violent Behavior, 59*, 101473. <https://doi.org/10.1016/j.avb.2020.101473>
- Skeem, J. L. & Mulvey, E. P. (2001). Psychopathy and community violence among civil psychiatric patients. *Journal of Consulting and Clinical Psychology, 69* (3), 358-374.

- Sleep, C. E., Weiss, B., Lynam, D. R., & Miller, J. D. (2019). An examination of the Triarchic Model of psychopathy's nomological network: A meta-analytic review. *Clinical Psychology Review, 71*, 1–26. <https://doi.org/10.1016/j.cpr.2019.04.005>
- Stewart-Williams, S. (2007). Altruism among kin vs. nonkin: Effects of cost of help and reciprocal exchange. *Evolution and Human Behavior, 28*(3), 193–198. <https://doi.org/10.1016/j.evolhumbehav.2007.01.002>
- Stone, A. A., Schneider, S., Smyth, J. M., Junghaenel, D. U., Couper, M. P., Wen, C., Mendez, M., Velasco, S., & Goldstein, S. (2024). A population-based investigation of participation rate and self-selection bias in momentary data capture and survey studies. *Current Psychology: A Journal for Diverse Perspectives on Diverse Psychological Issues, 43*(3), 2074–2090. <https://doi.org/10.1007/s12144-023-04426-2>
- van Lange, P. A. M., Schippers, M., & Balliet, D. (2011). Who volunteers in psychology experiments? An empirical review of prosocial motivation in volunteering. *Personality and Individual Differences, 51*(3), 279–284. <https://doi.org/10.1016/j.paid.2010.05.038>
- Vanags, P., Cutler, J., Kosse, F., & Lockwood, P. L. (2025). Greater income and financial well-being are associated with higher prosocial preferences and behaviors across 76 countries. *PNAS Nexus, 4*(2), pgae582. <https://doi.org/10.1093/pnasnexus/pgae582>
- Vekaria, K. M., Brethel-Haurwitz, K. M., Cardinale, E. M., Stoycos, S. A., & Marsh, A. A. (2017). Social discounting and distance perceptions in costly altruism. *Nature Human Behaviour, 1*(5), 0100. <https://doi.org/10.1038/s41562-017-0100>
- Virtanen, S., Latvala, A., Andershed, H., Lichtenstein, P., Tuvblad, C., Collins, O. F., Suvisaari, J., Larsson, H., & Lundström, S. (2022). Do psychopathic personality traits in childhood predict subsequent criminality and psychiatric outcomes over and above childhood

behavioral problems? *Journal of Criminal Justice*, 80, 101761.

<https://doi.org/10.1016/j.jcrimjus.2020.101761>

Winters, D. E., & Sakai, J. T. (2023). Affective theory of mind impairments underlying callous-unemotional traits and the role of cognitive control. *Cognition and Emotion*, 37(4), 696–713. <https://doi.org/10.1080/02699931.2023.2195154>

Winters, D. E., Spitz, J., Raymond, K., Natvig, C., Waller, R., Mikulich-Gilbertson, S. K., Schacht, J. P., & Sakai, J. T. (2025). Cognitive Control Difficulties Differentiate Callous-Unemotional Traits from Conduct Problems: A Pre-Registered Double-Blind Randomized Controlled Trial Analysis. *Child Psychiatry & Human Development*. <https://doi.org/10.1007/s10578-025-01869-5>

Wu, J., Balliet, D., Yuan, M., Li, W., Chen, Y., Jin, S., Luan, S., & Van Lange, P. A. M. (2025). Social class and prosociality: A meta-analytic review. *Psychological Bulletin*, 151(3), 285–321. <https://doi.org/10.1037/bul0000469>

Response to Reviewers:

REVIEWERS' COMMENTS:

Reviewer #1 (Remarks to the Author):

The authors have responded thoroughly to prior comments, and I appreciate the revisions made. This, I believe, will be a valuable contribution to the field.

One consideration for future publications is to place greater emphasis in the discussion section on how the findings align with, or diverge from, the contemporary literature. While additional citations were included, they currently appear somewhat appended rather than fully integrated. More explicit engagement with contemporary work would help situate the study in its nuanced context and highlight how these results inform interpretation and the design of future experiments.

At present, the discussion focuses primarily on points of agreement in the literature and the strengths of the study. A richer analysis that also acknowledges contrasting findings and explores their implications would elevate the discussion further.

One Steinbeis article I was thinking about - but not necessary to include - is here <https://doi.org/10.1016/j.copsyc.2017.08.012>

Thank you so much for this suggestion for our future publications. In light of both the reviewer's thoughtful input and the editor's request, we have revised the Discussion section to more explicitly engage with the cited literature. In particular, we now more clearly highlight:

- Where our results are consistent with existing findings,
- Where they diverge from prior studies,
- And how methodological or sample characteristics in our study may help explain these similarities, differences, and nuanced results

We present a few examples of our additions and edits below:

- On page 17, we revised the sentence beginning "Our results extend prior research on prosociality in psychopathy..." to more explicitly acknowledge that our findings align with prior evidence of reduced prosociality, while extending this work by identifying a specific mechanism and its link to meanness to explain antisocial behavior.
- On page 18, we revised the sentence beginning with "These findings add to evidence that the association between psychopathy and relevant outcome variables is not always linear..." to more directly connect with prior work and to emphasize the contribution of our sampling approach.
- Lastly, on page 19, we revised the paragraph comparing our findings to studies reporting smaller or null associations between psychopathy and social discounting. We added to this section to more clearly state the contrasting results and highlight key methodological differences - such as differences in task design, sampling strategy, and psychopathy

range - that may account for the divergent results. This supports the end of that paragraph, which was presented in the previous version, that contrasted our larger effect sizes with those in the prior literature and suggested plausible reasons for this difference.

We hope these changes address the reviewer's helpful comments and improve the clarity and contextualization of our findings within the existing literature.

Reviewer #2 (Remarks to the Author):

I am satisfied that the authors have addressed all the comments raised.

Thank you!

Reviewer #3 (Remarks to the Author):

All my comments have been thoroughly addressed by the authors. I look forward to reading the published paper!

Thank you for your helpful comments!